# H-Direct: Homeostasis-aware Direct Spike Encoding for Deep Spiking Neural Networks

## Abstract

Deep spiking neural networks (SNNs) have been expected to enable energy-efficient artificial intelligence as a next-generation artificial neural network. Recently, with the development of various algorithms, such as direct spike encoding, many applications have been successfully implemented in deep SNNs. Notably, most state-of-the-art deep SNNs have greatly improved their performance by adopting direct spike encoding, which expresses input information as discrete spikes, thereby exerting substantial influence. Despite the importance of the encoding, efficient encoding methods have not been studied. As the first attempt to our knowledge, we thoroughly analyzed the conventional direct encoding. Our analysis revealed that the existing encoding restricts the training performance and efficiency due to inappropriate encoding. To address this limitation by maintaining an appropriate encoding, we introduced a concept of homeostasis to the direct spike encoding. With this concept, we presented a homeostasis-aware direct spike encoding (H-Direct), which consists of dynamic feature encoding loss, adaptive threshold, and feature diversity loss. Our experimental results demonstrate that the proposed encoding achieves higher performance and efficiency compared to conventional direct encoding across several image classification datasets on various architectures. We have validated that brain-inspired algorithms have the potential to enhance the performance and efficiency of deep SNNs.

## 1 Introduction

Deep learning has shown remarkable performance in various artificial intelligence (AI) applications (Wu et al., 2022; Chang et al., 2024). However, such progress requires a lot of computation, which results in huge energy consumption. As AI technology utilizes deep learning advances to train larger models using more data, this energy consumption issue can no longer be overlooked. Deep neural networks (DNNs) with the latest performance consume a lot of energy not only for training but also for inference (McDonald et al., 2022; Desislavov et al., 2023). Thus, the energy problem has become the most urgent issue to be addressed for sustainable development and utilization of AI in our lives.

Neuromorphic computing, an emerging computing paradigm, has been expected to resolve this energy consumption issue of deep learning (Roy et al., 2019). By mimicking the human brain, it operates in an event-driven computing manner with spiking neural networks (SNNs), which leads to energy-efficient AI, especially on neuromorphic hardware (Ostrau et al., 2022). Recently, deep SNNs, which exploit the advantages of both DNNs and SNNs, have been expected to be the next-generation artificial neural networks for energy-efficient AI. Deep SNNs can simultaneously achieve high learning performance and low-energy operation by combining DNNs' synaptic topology and SNNs' asynchronous event-driven computing. The development of gradient-based learning algorithms, such as spatio-temporal backpropagation (STBP) with surrogate gradient (Wu et al., 2018; Neftci et al., 2019), has paved the way for the utilization of deep SNNs in various models and applications of DNNs (Guo et al., 2023; Su et al., 2023).

To fully leverage the advantages of deep SNNs, it is imperative to design an efficient neural coding scheme, which defines how information is represented with spikes. In particular, input spike encoding, which transforms the input signals into spike patterns, has significant effects on performance and efficiency. There are various types of input spike encoding, such as rate (Kim & Panda, 2021),

temporal (Park et al., 2020; Wei et al., 2023), and direct encoding (Rathi & Roy, 2021; Zheng et al., 2021; Guo et al., 2023). Among them, most state-of-the-art (SOTA) deep SNN models have adopted the direct encoding approach, which generates spikes in the first layer. Direct encoding learns encoding methods from data, which leads to superior performance over other encoding approaches. However, the existing direct encoding lacks consideration for stability and efficiency, which restricts the overall performance and efficiency of deep SNNs.

In this work, to overcome the aforementioned limitation, we first investigated conventional direct spike encoding. Following our analysis, we categorized the encoded spike channels into four types: over-fired, under-fired, dynamically selective, and persistent encoding, as shown in Fig. 1. With this categorization, we found that there were improperly encoded channels due to the inadequate firing rate, which limited the encoding layer's ability to express features from the input. Moreover, due to the lack of consideration for differences in features depending on input, the spike encoding for each channel was not optimized across inputs. Based on these analyses and inspired by the human brain, we introduced a concept of homeostasis into direct spike encoding for stable and appropriate encoding. With the concept of homeostasis, we propose a homeostasis-aware direct spike encoding, which is called H-Direct. This method enables stable and appropriate encoding by suppressing over- and under-firing while encouraging dynamic feature selection. The proposed approach consists of dynamic feature encoding loss, adaptive threshold, and feature diversity loss. Our comprehensive experiments demonstrated that the proposed homeostasis mechanism improved the training performance and efficiency of deep SNNs on various datasets, models, and training algorithms, which showed effectiveness and versatility of the proposed approach.

## 2 RELATED WORK AND PRELIMINARIES

### 2.1 DEEP SPIKING NEURAL NETWORKS

SNNs, which mimic the operation of the brain, have been considered the next generation of artificial neural networks (Maass, 1997). SNNs propagate information using spikes through a network of neurons and synapses, enabling energy-efficient operations with asynchronous event-driven computing. Deep SNNs can simultaneously achieve high learning ability and energy-efficient operation by combining DNNs' synaptic topology and SNNs' event-based operation (Tavanaei et al., 2019). Leaky integrate-and-fire (LIF) neurons are widely used in deep SNNs due to their low computational cost. The integration process of LIF neurons can be described as

$$u_i^l[t] = 1/\tau(v_i^l[t-1] + \sum_j w_{ij}s_j^{l-1}[t]), \tag{1}$$

where $u$, $v$, $w$, and $s$ indicate the neuron's internal state, called membrane potential, intermediate state, synaptic weight, and input spike, respectively. The layer index is $l$, and the neuron indices are $i$ and $j$. The time constant and time step are represented in $\tau$ and $t$, respectively. A spike is generated when the membrane potential exceeds the threshold as

$$s_i^l[t] = H(u_i^l[t] - V_{\text{th}}), \tag{2}$$

where $H$ and $V_{\text{th}}$ are the Heaviside step function and a threshold voltage, respectively. When a neuron fires a spike, its membrane potential is reset through the intermediate state, which can be stated as

$$v_i^l[t] = (u_i^l[t] - s_i^l[t])s_i^l[t] + u_i^l[t](1 - s_i^l[t]). \tag{3}$$

Recently, various deep learning applications and models have been implemented with deep SNNs, such as image classification (Fang et al., 2021; Hu et al., 2021; 2024), multi-object detection (Kim et al., 2020a;b), and Transformer (Zhou et al., 2023; Yao et al., 2024). Most of these SOTA deep SNN models adopted STBP with surrogate gradient (Wu et al., 2018), threshold-dependent batch normalization (tdBN) (Zheng et al., 2021), and direct spike encoding (Rathi & Roy, 2021; Wu et al., 2021; Zheng et al., 2021; Deng et al., 2022; Guo et al., 2022b; 2023). The training algorithms and spike encoding methods have greatly contributed to the successful implementation of deep SNNs, but there are still training performance gaps between DNNs and deep SNNs. To narrow these gaps, many studies have been conducted, including training algorithms (Wu et al., 2019; Rathi & Roy, 2021), and resolving gradient mismatching caused by surrogate gradient (Li et al., 2021; Lian et al., 2023). However, insufficient attention has been given to research on improving spike encoding.

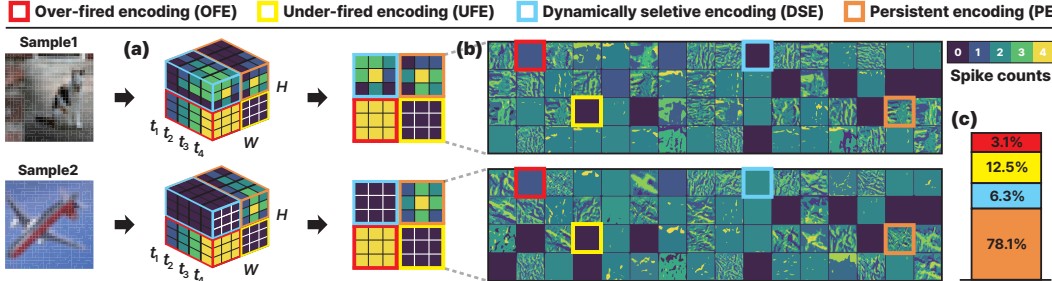

Figure 1: (a) The direct encoding converts input data into spikes over several time steps, which are then accumulated over time (i.e. along with channels), producing output spike features. (b) Each feature can be categorized into over-fired encoding (OFE), under-fired encoding (UFE), dynamically selective encoding (DSE), and persistent encoding (PE). Each box represents a single example of the encodings. (c) Proportions of each categorized encoding.

## 2.2 SPIKE ENCODING

Spike encoding determines how input signals are expressed as spikes, a form of information that can be processed by SNNs (Auge et al., 2021). Since the encoded spikes convey information from input data, the encoding process significantly impacts the performance and efficiency of SNNs. Previous research has proposed various encoding schemes such as rate (Van Rullen & Thorpe, 2001) and temporal encoding, including phase (Kim et al., 2018; Park et al., 2019), and time-to-first-spike (TTFS) (Comsa et al., 2020; Han & Roy, 2020; Park et al., 2020; Park & Yoon, 2021) for the efficient processing of input in deep SNNs. However, these encoding methods restricted the training performance of deep SNNs due to loss of input information. To address this, SOTA deep SNN models have commonly adopted direct encoding (Guo et al., 2023; Yao et al., 2024). In this approach, the first layer is designated as the encoding layer, which is responsible for learning an encoding method from data. This encoding layer is trained in an end-to-end manner, simultaneously with the other layers in the model. Direct encoding has been experimentally validated to improve the training of deep SNNs across various datasets and models (Rathi & Roy, 2021; Zheng et al., 2021; Deng et al., 2022; Guo et al., 2022b; 2023; Yao et al., 2024). However, due to a lack of consideration of SNN characteristics, the encoding has constrained the training capability of deep SNNs. In the recent study (Qiu et al., 2024), an attention mechanism was applied to the encoding layer of direct encoding, achieving SOTA accuracy. However, this approach imposes additional overhead compared to conventional direct encoding due to the need for an extra attention layer in the encoding process.

## 2.3 HOMEOSTASIS IN SNNs

Homeostasis is crucial for the appropriate functionality of biological systems by maintaining internal stability (Fernandes & Carvalho, 2016). The absence of homeostasis makes the system unstable, which results in degradation of information processing ability and efficiency (Miller & MacKay, 1994; Abbott & Nelson, 2000; Abraham et al., 2002). Thus, the learning process of the neural network should incorporate homeostatic mechanisms to maintain appropriate firing rate (Turrigiano & Nelson, 2004), such as synaptic scaling observed *in vitro* (Turrigiano et al., 1998) and *in vivo* (Keck et al., 2013). Few studies have introduced the biological efficiency of homeostasis on SNNs. In (Diehl & Cook, 2015), homeostasis was introduced through an adaptive threshold to improve training performance. However, it had limitations as it could not be applied to deep SNNs. Another recent study showed adversarial robustness through homeostasis using an adaptive threshold (Geng & Li, 2023). Nonetheless, this study focused on the stability of the neural network and failed to show the possibility of improving information processing. Hence, it is imperative to investigate methods for enhancing the training performance and efficiency of deep SNNs with homeostasis.

## 3 ANALYSIS OF CONVENTIONAL DIRECT ENCODING

To improve spike encoding, we analyzed the conventional direct encoding widely used in deep SNNs. To clearly observe the impact of direct encoding, we trained deep SNNs with STBP using surrogate gradient and tdBN, which are the current standard training approach. Direct encoding

employs the first layer of deep SNNs as the encoding layer, which usually consists of synaptic connection (e.g., convolution), normalization (e.g., batch normalization), and encoding neurons (Fang et al., 2021; Zheng et al., 2021; Zhou et al., 2023; Yao et al., 2024). The direct encoding extracts features from input data and encodes them into spikes in a channel-wise manner according to the time step $t$, as shown in Fig. 1-(a). The accumulated encoded spikes during a total time step $T$ are depicted in Fig. 1-(b). According to the encoding aspect, we found that the encoded features (channels) can be categorized into four types: over-fired, under-fired, persistent, and dynamically selective encoding.

Over- and under-fired encoding (OFE and UFE, respectively) are caused by an inadequate firing rate of encoding neurons (red and yellow boxes in Fig. 1-(b)). These inappropriate encoded channels cannot encode any features because all neurons in the channel are fired with the same value. As in many other studies (Hwang et al., 2020; 2021), such inappropriate encoding should be avoided since it limits the training ability of deep SNNs. Persistent encoding (PE) consistently converts the extracted features into spikes regardless of input, as in DNNs (orange box in Fig. 1-(b)). In this case, the generation of encoded spikes in every input results in inefficient deep SNNs that rely on event-driven computing. Lastly, dynamically selective encoding (DSE) generates encoding spikes depending on the input (blue box in Fig. 1-(b)). In this encoding, only essential features of the input are encoded into spikes. As illustrated in Fig. 1-(b), the feature corresponding to the blue box is encoded in Sample2 but not in Sample1. Such selective encoding according to inputs can reduce the number of spikes, thereby improving the energy efficiency of deep SNNs.

The proportions of the four types of encoding for Fig. 1-(b) are shown in Fig. 1-(c). PE has the highest proportions, while there are also inappropriately encoded channels (OFE and UFE). This suggests that the conventional direct encoding needs improvement to achieve a proper encoding rate with more selective encoding features, ultimately resulting in more efficient deep SNNs.

## 4 HOMEOSTASIS-AWARE DIRECT SPIKE ENCODING

In Sec. 3, we observed improper encoding in the conventional direct encoding, which hindered improvement in performance and efficiency of deep SNNs. To enhance encoding stability, we introduce homeostasis to spike encoding with the following definition: "*the homeostasis of spike encoding is the property that maintains appropriate encoding regardless of input*". This can be accomplished by the three factors: (i) preventing inappropriate firing rate of encoding neurons, (ii) encoding only essential features depending on input, and (iii) encoding diverse features by enhancing the utilization of model capacity. To facilitate these in deep SNNs, we propose a novel direct spike encoding called H-Direct, which consists of dynamic feature encoding (DFE) loss, adaptive threshold (AT), and feature diversity (FD) loss. Detailed explanations for each method are provided in the following sections.

### 4.1 DYNAMIC FEATURE ENCODING LOSS

Conventional direct spike encoding, which has been widely used in deep SNNs, mostly exploits the same structure as "*Conv-BN-Neuron*" regardless of the model architectures, as shown in Fig. 2 (Zheng et al., 2021; Deng et al., 2022; Guo et al., 2022a; 2023). Thus, since the output of batchnormalization (BN) is used as the input of the encoding neuron, the parameters of BN significantly impact spike encoding. For example, if the influence of the shift parameter ($\beta$) is greater than the scale parameter ($\gamma$) of BN, the deviation decreases depending on the input, strengthening the deterministic behavior of the encoding. In the opposite case, different encoding patterns appear frequently depending on the input. Based on this intuition, we found that the ratio of the scale parameter and shift parameter of BN for each channel ($\beta_c/\gamma_c$) is closely related to the type of encoding channel categorized in Sec. 3. As shown in Fig. 2-(a) (at epoch 1), each encoding type has a distinguished distribution according to the ratio. If this value is excessively positive or negative compared to the threshold, the firing of encoding neurons is excessively promoted or suppressed, resulting in OFE or UFE, respectively. Moreover, if the ratio is moderately positive near the threshold, the channel has a high probability of firing spikes and thus acts as a PE that encodes spikes for all inputs. When the ratio is adequately negative or positive, the channel operates as a DSE whose encoding is determined by the input.

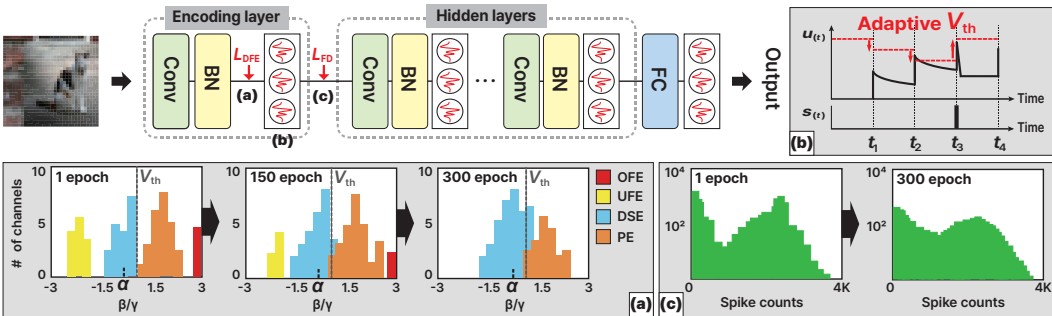

Figure 2: An overview of our proposed method, H-Direct, which achieves this with three main modules: (a) dynamic feature encoding loss ($L_{\text{DFE}}$), which suppresses OFE and UFE while increasing DSE and PE, enabling us to dynamically select the essential features to encode as spikes based on the input, (b) adaptive threshold that triggers the firing of non-encoded channels, and (c) histograms showing the spike count distributions of encoded channels during training. As illustrated in (c), the application of feature diversity loss ($L_{\text{FD}}$) leads to a more dispersed distribution, thereby enhancing encoding performance by encouraging diverse features.

With these observations, we propose DFE loss that makes this ratio a certain value in the distribution of DSE to prohibit inappropriate encodings and improve encoding efficiency. We define DFE loss as follows:

$$L_{\text{DFE}} = \sum_c \left\| \frac{\beta_c}{\gamma_c + \epsilon} - \alpha \right\|_2, \tag{4}$$

where $c$, $\alpha$, and $\epsilon$ are the channel index, the target value of the ratio ($\beta/\gamma$), and a small positive number for the numerical stability, respectively. This loss encourages the ratio to be trained to the target value $\alpha$ at which DSE channels are likely to occur. The gradients of each parameter for the loss are presented as

$$\frac{\partial L_{\text{DFE}}}{\partial \beta_c} = \frac{\chi_c}{\|\chi_c - \alpha\|_2} \frac{1}{(\gamma_c + \epsilon)}, \quad \frac{\partial L_{\text{DFE}}}{\partial \gamma_c} = -\frac{\chi_c^2}{\|\chi_c - \alpha\|_2} \frac{1}{(\gamma_c + \epsilon)}, \tag{5}$$

where $\chi_c = \frac{\beta_c}{(\gamma_c + \epsilon)}$. The more detailed derivation is in Sec. A.3.

When DFE is applied, the encoding neurons' firing rate of each channel is maintained appropriately as training progresses. Accordingly, unsuitable encoding channels (UFE and OFE) were eliminated, as shown in Fig. 2-(a). Furthermore, a larger proportion of encoding channels transitions to DSE, which improves the efficiency of encoding.

## 4.2 Adaptive Threshold in Encoding Neurons

While DFE can eliminate encoding channels that lead to inappropriate firing, this may negatively impact training performance. Specifically, when the membrane potential of neurons accumulates to substantial negative values, DFE needs to exert significant effort to induce their firing, which can degrade the performance. To overcome this, we introduce an adaptive threshold in encoding neurons. The proposed threshold is adjusted channel-wise to ensure computational efficiency and precise adjustment. The adaptive increase in threshold may suppress neurons with lower firing rates within the same channel, consequently deteriorating the encoding performance. Thus, we propose an asymmetric adaptive threshold that promotes the firing of non-encoded channels to improve homeostasis while having less influence on other encoding channels. The proposed method can be expressed as follows:

$$V_{\text{th},c}(t) = \begin{cases} \eta V_{\text{th},c}(t-1) & \text{if} \quad \sum_{\{i \in Channel_c\}} s_i[t] = 0 \\ V_{\text{th},c}(0) & \text{otherwise} \end{cases}, \tag{6}$$

where $c$, $\eta$, and $V_{\text{th}}(0)$ denote channel index, adjust rate, and initial threshold, respectively. As depicted in Fig. 2-(b), this adjustment can be cumulative, but once firing occurs, the threshold is restored to its initial value for subsequent time steps. This method enables the encoding layer to fully utilize its potential by promoting the encoding of non-firing channels.

### 4.3 FEATURE DIVERSITY LOSS

As discussed in the homeostasis of encoding, another factor for stable encoding is diversity in encoding features, which can be achieved by maximizing the utilization of model capacity. By extracting diverse features from the input data and encoding them into spikes, it is possible to achieve stable encoding that consistently generates appropriate spike patterns. From this perspective, DFE, which primarily focuses on efficiency and minimal encoding, struggles to induce diverse feature representations. Thus, to achieve more stable and effective spike encoding, we propose an FD loss that encourages diverse feature encoding by maximizing the entropy of features. However, while calculating feature entropy, there was distortion in the distribution of features since the feature space was severely undersampled relative to the dimensionality. To address this, we used the accumulated spike distribution of neurons in each channel as a surrogate for the feature distribution and fitted it to the probability density function (PDF). The general form of FD loss can be represented as

$$L_{\text{FD}} = -\sum_k p(x_k) \log p(x_k), \tag{7}$$

where $x_k$ and $p(x_k)$ denote the feature and PDF, respectively. In order for the proposed loss to be compatible with a gradient-based training algorithm, the PDF must be differentiable. Thus, we used a normal distribution $\mathcal{N}(\mu, \sigma)$ as the PDF, where $\mu$ and $\sigma$ are the mean and standard deviation of accumulated spikes. The gradient of the encoding layer for the feature diversity loss can be stated as

$$\frac{\partial L_{\text{FD}}}{\partial W} \approx \sum_k -\log(p(x_k) - 1)p'(x_k) \sum_t I[t]/\tau, \tag{8}$$

where $I$ is the input. The more detailed derivation is in Sec. A.2. As shown in Fig. 2-(c), this method improves encoding performance by encouraging diverse feature encoding.

Our overall loss function including cross-entropy loss for the image classification is as follows:

$$L = \lambda_{\text{CE}} L_{\text{CE}} + \lambda_{\text{FD}} L_{\text{FD}} + \lambda_{\text{DFE}} L_{\text{DFE}}, \tag{9}$$

where $\lambda_{\text{CE}}$, $\lambda_{\text{FD}}$, and $\lambda_{\text{DEF}}$ denote the weights factors of $L_{\text{CE}}$, $L_{\text{FD}}$, and $L_{\text{DEF}}$, respectively.

## 5 EXPERIMENTS

To evaluate the effectiveness of our proposed encoding approach, we conducted extensive experiments with typical model architectures (i.e., VGG16, ResNet19, and ResNet20) on various datasets, such as static image datasets (CIFAR10, CIFAR100, and ImageNet) and neuromorphic dataset (CIFAR10-DVS). We use the training algorithm of the STBP-tdBN (Zheng et al., 2021) (a threshold-dependent batch normalization method based on the spatio-temporal backpropagation) as our baseline. Furthermore, we applied our proposed encoding approach to other training algorithms, such as IM-loss (Guo et al., 2022a) and RMP-loss (Guo et al., 2023). For all experiments, average scores over four independent runs are reported for each configuration to ensure a fair comparison. Following the conventions, we use LIF neurons with a soft reset (Eq. 3) and set the time step to four. For more details about experimental setup and implementation, please refer to Sec. A.4.

### 5.1 QUANTITATIVE ANALYSIS

**Comparison with Baseline.** We start by analyzing the effect of our proposed encoding method by visualizing the spike feature maps and comparing the proportions of the following four encoding categories: i.e., OFE, UFE, DSE, and PE. In Fig. 3, we observe that applying our proposed H-Direct notably reduces the OFE and UFE rates, maintaining an appropriate firing rate as we intended. In Fig. 3-(a), we provide examples of encoded feature maps between only two samples, differently color-coded according to their encoding categories. Further, in Fig. 3-(b), we provide the averaged proportion of each encoding category over all test sets. Notably, we observe that the DSE rate increases to 62.9% (12.9%↑) while the PE rate decreases to about 37% (2.0%↓).

Further, as shown in Tab. 1, we measured (i) the classification accuracy, (ii) the number of total spikes (which refers to the number of spikes that are fired by neurons across all layers), and (iii) the number of encoded spikes (which refer to the number of spikes that are fired in the encoding layer). We

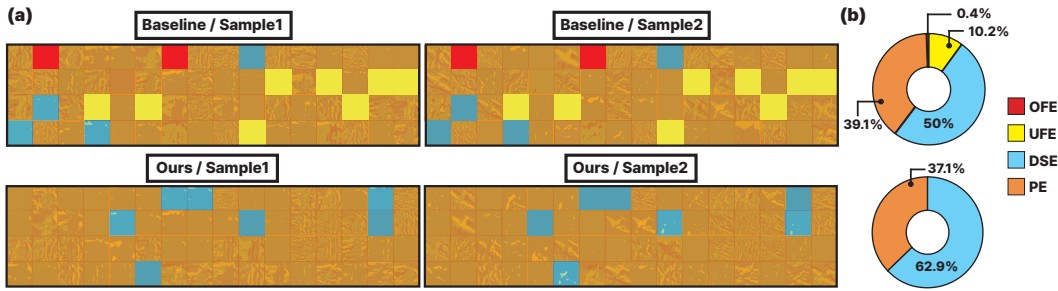

Figure 3: (a) Comparison of encoded feature maps between baseline (top) and ours (bottom). We also highlight grids with different colors according to their encoding categories (i.e., red: OFE, yellow: UFE, blue: DSE, and orange: PE). (b) The average proportions of each encoding category over all test sets. Note that we use VGG16 on CIFAR10.

Table 1: Given four different datasets (CIFAR10, CIFAR100, CIFAR10-DVS, and ImageNet), we compare ours with the baseline training algorithm (STBP-tdBN) based on various architectures, including VGG16 and ResNet-based models.

| | Accuracy (%) | | | # of *All* Spikes (in k) | | | # of *Encoded* Spikes (in k) | | |
|---|---|---|---|---|---|---|---|---|---|
| | Baseline | Ours | Δ | Baseline | Ours | Δ | Baseline | Ours | Δ |
| *Data: CIFAR10* | | | | | | | | | |
| VGG16 | 93.47±0.14 | 93.67±0.06 | +0.21 | 148±8.0 | 144±6.0 | -3.00% | 59±2.00 | 52±0.8 | -12.00% |
| ResNet19 | 95.61±0.03 | 95.72±0.18 | +0.12 | 825±12.0 | 781±11.0 | -5.30% | 190±0.04 | 188±0.8 | -1.05% |
| ResNet20 | 94.99±0.02 | 95.09±0.04 | +0.11 | 480±11.0 | 463±5.0 | -3.54% | 92±1.20 | 83±0.9 | -9.78% |
| *Data: CIFAR100* | | | | | | | | | |
| VGG16 | 69.03±0.13 | 69.29±0.05 | +0.38 | 160±0.6 | 151±1.0 | -6.00% | 64±1.00 | 57±0.9 | -11.00% |
| ResNet19 | 76.86±0.05 | 77.07±0.10 | +0.23 | 1003±8.0 | 987±7.0 | -1.60% | 222±2.00 | 217±0.6 | -2.30% |
| ResNet20 | 74.92±0.03 | 75.13±0.12 | +0.28 | 629±5.0 | 624±0.8 | -0.78% | 121±0.60 | 118±0.5 | -2.29% |
| *Data: CIFAR10-DVS* | | | | | | | | | |
| VGG16 | 75.10±0.16 | 76.15±0.62 | +1.40 | 413±1.2 | 273±1.8 | -33.90% | 146±0.50 | 18±0.2 | -87.74% |
| *Data: ImageNet* | | | | | | | | | |
| ResNet18 | 64.07±0.08 | 64.30±0.03 | +0.36 | 2175±15.0 | 2051±7.0 | -5.70% | 872±8.00 | 722±5.0 | -17.20% |
| ResNet34 | 68.29±0.06 | 68.43±0.02 | +0.21 | 3079±9.0 | 2908±17.0 | -5.55% | 840±9.00 | 608±6.0 | -27.62% |

compare ours with the baseline training algorithm (i.e., STBP-tdBN) with different architectures and datasets, including CIFAR10, CIFAR100, CIFAR10-DVS, and ImageNet. We observe that applying H-Direct consistently improves the overall classification accuracy in all experiments, significantly reducing the number of fired spikes (0.78–33.90%↓ for all spikes and 1.05–87.74%↓ for encoded spikes). Notably, this trend is more significant with the neuromorphic dataset, i.e., CIFAR10-DVS. As shown in supplemental Tab. 7, we observe that the conventional direct encoding results in a dominant proportion of PE. In contrast, with H-Direct, the proportion of DSE increases to 23.8%, offering significant efficiency improvements.

**Noise Robustness.** The model's robustness against noise is crucial for deploying deep SNNs on various real-world neuromorphic devices (Park et al., 2021; Yang et al., 2022). To evaluate the effect of our encoding method on noise robustness, we measure the classification accuracy in terms of different levels of two noise types: (i) input and (ii) integration noise. The former occurs before the encoding layer, while the latter occurs in encoding neuron's membrane potential. Note that we applied Gaussian noise $\mathcal{N}(0, \sigma)$

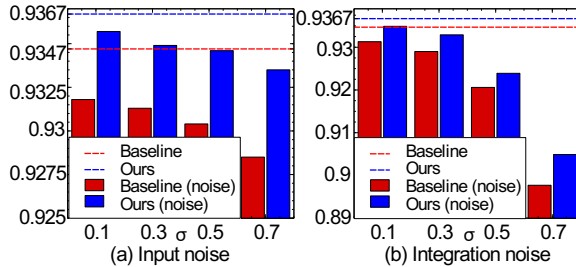

Figure 4: Classification accuracy comparison against different levels of (a) input and (b) integration noise.

into input and membrane potential ($u$ in Eq. 1). Given VGG16 architecture, we compare the accuracy between ours and our baseline. As shown in Fig. 4, our encoding approach generally provides

Table 2: Comparison with current state-of-the-art approaches in terms of classification accuracy and the number of spikes. We experimented with different datasets and architectures. * denotes our implementation.

| Datasets | Architectures | Methods | Time steps | Accuracy (in %) | # of Spikes (in k) |
|---|---|---|---|---|---|
| CIFAR10 | VGG16 | Diet-SNN (Rathi & Roy, 2021) | 5 | 92.70 | - |
| | | tdBN (Zheng et al., 2021)* | 4 | 93.47±0.14 | 148±8 |
| | | tdBN + H-Direct | 4 | 93.67±0.06 (0.20↑) | 144±6 (4↓) |
| | | RMP (Guo et al., 2023)* | 4 | 93.59±0.03 | 155±2 |
| | | RMP + H-Direct | 4 | 93.69±0.05 (0.10↑) | 133±7 (22↓) |
| | | IM (Guo et al., 2022a)* | 4 | 93.73±0.03 | 142±1 |
| | | IM + H-Direct | 4 | 93.89±0.01 (0.16↑) | 137±6 (5↓) |
| | ResNet19 | TET (Deng et al., 2022) | 4 | 94.44±0.08 | - |
| | | TAB (Jiang et al., 2024) | 4 | 94.76 | - |
| | | LOCALZO (Mukhoty et al., 2023) | 4 | 94.89 | - |
| | | RMP (Guo et al., 2023)* | 4 | 95.23±0.13 | 963±29 |
| | | RMP + H-Direct | 4 | 95.23±0.13 (0.00↑) | 955±26 (8↓) |
| | | tdBN (Zheng et al., 2021)* | 4 | 95.61±0.03 | 825±12 |
| | | tdBN + H-Direct | 4 | 95.72±0.18 (0.11↑) | 781±11 (44↓) |
| | | IM (Guo et al., 2022a)* | 4 | 95.76±0.06 | 1116±13 |
| | | IM + H-Direct | 4 | 95.78±0.14 (0.02↑) | 1075±26 (41↓) |
| | ResNet20 | Diet-SNN (Rathi & Roy, 2021) | 5 | 91.78 | - |
| | | RMP (Guo et al., 2023)* | 4 | 94.77±0.07 | 619±14 |
| | | RMP + H-Direct | 4 | 94.79±0.10 (0.02↑) | 586±20 (33↓) |
| | | tdBN (Zheng et al., 2021)* | 4 | 94.99±0.02 | 480±11 |
| | | tdBN + H-Direct | 4 | 95.09±0.04 (0.10↑) | 463±5 (17↓) |
| | | IM (Guo et al., 2022a)* | 4 | 95.16±0.13 | 752±18 |
| | | IM + H-Direct | 4 | 95.23±0.06 (0.07↑) | 646±7 (106↓) |
| CIFAR100 | VGG16 | Diet-SNN (Rathi & Roy, 2021) | 5 | 69.97 | - |
| | | tdBN (Zheng et al., 2021)* | 4 | 69.03±0.13 | 160±0.6 |
| | | tdBN + H-Direct | 4 | 69.29±0.05 (0.26↑) | 151±1 (9↓) |
| | | RMP (Guo et al., 2023)* | 4 | 69.35±0.13 | 715±5 |
| | | RMP + H-Direct | 4 | 69.49±0.14 (0.14↑) | 157±2 (558↓) |
| | | IM (Guo et al., 2022a)* | 4 | 69.68±0.05 | 174±3 |
| | | IM + H-Direct | 4 | 69.74±0.17 (0.06↑) | 161±2 (13↓) |
| | ResNet19 | LOCALZO (Mukhoty et al., 2023) | 4 | 74.13 | - |
| | | TET (Deng et al., 2022) | 4 | 74.47±0.15 | - |
| | | TAB (Jiang et al., 2024) | 4 | 76.81 | - |
| | | RMP (Guo et al., 2023)* | 4 | 76.13±0.08 | 1147±13 |
| | | RMP + H-Direct | 4 | 76.43±0.07 (0.3↑) | 1104±8 (43↓) |
| | | tdBN (Zheng et al., 2021)* | 4 | 76.86±0.05 | 1003±8 |
| | | tdBN + H-Direct | 4 | 77.07±0.10 (0.21↑) | 987±7 (16↓) |
| | | IM (Guo et al., 2022a)* | 4 | 76.94±0.11 | 1309±11 |
| | | IM + H-Direct | 4 | 77.15±0.23 (0.21↑) | 1284±9 (25↓) |
| | ResNet20 | Diet-SNN (Rathi & Roy, 2021) | 5 | 64.07 | - |
| | | tdBN (Zheng et al., 2021)* | 4 | 74.92±0.03 | 629±5 |
| | | tdBN + H-Direct | 4 | 75.13±0.12 (0.21↑) | 624±1 (5↓) |
| | | RMP (Guo et al., 2023)* | 4 | 74.38±0.16 | 715±5 |
| | | RMP + H-Direct | 4 | 74.60±0.25 (0.22↑) | 703±7 (12↓) |
| | | IM (Guo et al., 2022a)* | 4 | 74.94±0.16 | 797±8 |
| | | IM + H-Direct | 4 | 75.41±0.08 (0.47↑) | 764±2 (33↓) |
| ImageNet | ResNet18 | RMP (Guo et al., 2023) | 4 | 63.03±0.07 | - |
| | | tdBN (Zheng et al., 2021)* | 4 | 64.07±0.08 | 2175±15 |
| | | tdBN + H-Direct | 4 | 64.30±0.03 (0.23↑) | 2051±7 (124↓) |
| | ResNet34 | RMP (Guo et al., 2023) | 4 | 65.17±0.07 | - |
| | | IM (Guo et al., 2022a) | 4 | 67.43±0.11 | - |
| | | TAB (Jiang et al., 2024) | 4 | 67.78 | - |
| | | tdBN (Zheng et al., 2021)* | 4 | 68.29±0.06 | 3079±9 |
| | | tdBN + H-Direct | 4 | 68.43±0.02 (0.14↑) | 2908±17 (171↓) |

better robustness against both types of noise. This suggests that the homeostasis-aware encoding approach may enhance overall training performance and efficiency, while also improving robustness to noise.

Table 3: Comparisons with the current state-of-the-art approaches on the neuromorphic dataset, i.e., CIFAR10-DVS. We report the classification accuracy (in %) and the number of spikes (in k).

| Architectures | Methods | Time steps | Accuracy (in %) | # of *All* Spikes (in k) |
|---|---|---|---|---|
| ResNet19 | STBP-tdBN (Zheng et al., 2021) | 10 | 67.80 | - |
| ResNet20 | RMP (Guo et al., 2023) | 10 | 75.60±0.30 | - |
| VGG16 | STBP-tdBN (Zheng et al., 2021) | 4 | 75.10±0.08 | 413±1 |
| | Ours | 4 | 76.15±0.31 (1.05↑) | 273±2 (140↓) |

**Comparison with Current SOTA Approaches.** Further, as shown in Tab. 2, we conducted a more extensive comparison with the current SOTA approaches, including RMP (Guo et al., 2023), LO-CALZO (Mukhoty et al., 2023), TAB (Jiang et al., 2024), IM (Guo et al., 2022a), Diet-SNN (Rathi & Roy, 2021), tdBN (Zheng et al., 2021), and TET (Deng et al., 2022). Similar to our previous analysis, we report the overall classification accuracy and the number of spikes. Note that an ideal model may have higher classification accuracy with fewer spikes. Following conventions, we conducted experiments with various architectures, i.e., VGG16, ResNet19, and ResNet20 for CIFAR10 and CIFAR100 datasets while ResNet18 and ResNet34 for ImageNet. We observe that our proposed method, which is built upon the baseline (STBP-tdBN), shows promising scores that are matched or better than the current SOTA approaches. Further, we applied our H-Direct approach to other alternatives (i.e., IM (Guo et al., 2022a) and RMP (Guo et al., 2023)), which consistently provides improved accuracy and efficiency. Moreover, we conducted experiments with the neuromorphic dataset, CIFAR10-DVS, as shown in Tab. 3. We observed that a model with our encoding approach achieved the SOTA-level performance with better efficiency. This may suggest that our method effectively improves performance and efficiency across diverse datasets, architectures, and methods.

## 5.2 ABLATION STUDIES

We further conducted ablation studies to demonstrate the individual contribution of three proposed main components: (i) dynamic feature encoding (DFE) loss, (ii) adaptive threshold (AT), and feature diversity (FD) loss. As summarized in Tab. 4, we compare the variant of our models in terms of the accuracy and the number of all spikes and encoded spikes. Note that our experiments are based on the CIFAR10 dataset with VGG16 and ResNet20. We observe in Tab. 4 that applying DFE only significantly improves the model's efficiency, potentially due to selective encoding. Further, applying other building blocks, AT and FD, enhances the overall classification accuracy with marginal sacrificing efficiency. This is due to the fact that DFE generally improves efficiency by leaving only neurons with appropriate firing, while AT and FD provide tension by encouraging feature diversity.

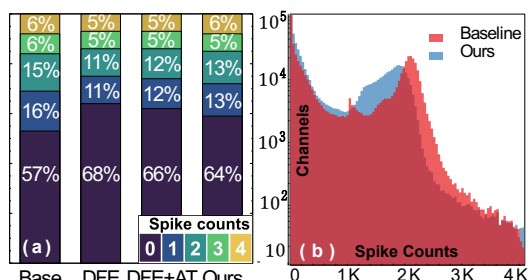

Figure 5: (a) Proportion and (b) distributions of spike counts of encoding neurons and channels, respectively (VGG16, CIFAR10). The y-axis of (b) is set to a log scale.

To understand in more detail the impact of the proposed method on spike encoding, we investigated the encoding patterns of neurons and channels within the encoding layer. The spike count proportion of the encoding neurons for each ablation is shown in Fig. 5-(a). As can be seen in the figure, DFE reduces the proportion of neurons with low spike counts (one or two) and increases the proportion of non-spiking neurons. This shows that dynamic selectivity only affects low-firing neurons. DFE+AT assists in the firing of non-firing neurons, slightly increasing the proportion of neurons with one spike count. However, since DFE aims to maintain an appropriate firing rate, it does not cause significant changes. Consequently, ours, with the addition of FD, makes the spike counts distribution of DFE+AT more diverse, contributing to enhanced feature diversity and improved performance.

We also analyzed the effect of the proposed method at the feature level with spike count distributions of each channel, as shown in Figs. 5-(b), 6, and 7. When DFE is applied, low-firing features are

Table 4: Ablation study results to evaluate the individual contributions of our components, i.e., DFE (dynamic feature encoding loss), AT (adaptive threshold), and FD (feature diversity loss). Data: CIFAR10, Baseline: STBP-tdBN.

| Architectures | Methods | Accuracy (in %) | # of Total Spikes (in k) | # of Encoded Spikes (in k) |
|---|---|---|---|---|
| VGG16 | Baseline | 93.47±0.14 | 148±8.0 | 59±2.0 |
| | w/ DFE only | 93.53±0.11 | 135±4.0 | 48±1.0 |
| | w/ DFE+AT | 93.55±0.14 | 135±7.8 | 52±5.9 |
| | w/ DFE+AT+FD (ours) | 93.67±0.06 | 144±6.0 | 52±0.8 |
| ResNet20 | Baseline | 94.99±0.02 | 480±11.0 | 92±1.0 |
| | w/ DFE only | 94.82±0.03 | 444±5.0 | 33±0.7 |
| | w/ DFE+AT | 94.88±0.05 | 455±8.3 | 34±0.7 |
| | w/ DFE+AT+FD (ours) | 95.08±0.05 | 460±8.5 | 39±0.5 |

Table 5: Ablation studies to compare variants of our method with and without AT (adaptive threshold), FD (feature diversity loss), and DFE (dynamic feature encoding loss). Data: CIFAR10, Baseline: STBP-tdBN.

| Model | Methods | Cross-correlation | # of Channels | Proportions (in %) | | | |
|---|---|---|---|---|---|---|---|
| | | | | OFE | UFE | DSE | PE |
| VGG16 | Baseline | 0.231 | 51.55 | 0.39 | 10.16 | 50.00 | 39.06 |
| | w/ DFE | 0.203 | 56.77 | 0.00 | 0.00 | 66.80 | 33.20 |
| | w/ DFE+AT | 0.230 | 56.99 | 0.00 | 0.00 | 63.67 | 36.33 |
| | w/ DFE+AT+FD (ours) | 0.218 | **59.51** | 0.00 | 0.00 | 62.89 | 37.11 |
| ResNet20 | Baseline | 0.433 | 62.07 | 0.00 | 2.34 | 47.66 | 50.00 |
| | w/ DFE | 0.221 | 62.83 | 0.00 | 0.00 | 73.83 | 26.17 |
| | w/ DFE+AT | 0.241 | 63.25 | 0.00 | 0.00 | 69.53 | 30.47 |
| | w/ DFE+AT+FD (ours) | 0.237 | **63.27** | 0.00 | 0.00 | 73.44 | 26.56 |

suppressed (Fig. 6-(b)). The model with DFE+AT encourages features to be encoded with small spikes (i.e., <1k) (Fig. 6-(c)). Our method promotes low-firing features while preventing high-firing features, thereby achieving both diversity and efficiency in encoding (Fig. 5-(b)).

For further analysis, we measured the correlation between features (Jin et al., 2020), the average number of channels used for encoding, and the proportions of each channel type, which are presented in Tab. 5. DFE shows the lowest cross-correlation, but this is due to insufficient encoding that does not fully utilize the encoding channels. The proportion of DSE is the highest compared to the others. In DFE+AT, the firing of non-encoded channels is promoted, which leads to an increase in encoded channels. Lastly, our method demonstrates the highest utilization of encoding channels. Despite the increase in the number of encoding channels, it effectively reduces redundant features, thereby decreasing cross-correlation compared to DFE+AT. In the case of ResNet20, the overall trend is similar to that of VGG16. However, due to structural differences such as residual connections, ResNet20 utilizes more channels for encoding, and the proportion of DSE is also higher compared to VGG16.

# 6 CONCLUSION

In this work, we proposed a novel homeostasis-aware direct spike encoding called H-Direct, which we demonstrated with extensive experiments that applying H-Direct is indeed able to improve the efficiency and efficacy of deep SNNs together. Our work starts with a thorough analysis of conventional direct encoding approaches, which led to the conceptualization of brain-inspired homeostasis in spike encoding. To offer homeostasis-aware direct spike encoding, we proposed the following three main components: (i) dynamic feature encoding loss, (ii) adaptive threshold, and (iii) feature diversity loss. Our extensive experiments showed that our method could improve the efficiency and stability of spike encoding, enhancing the overall training performance and efficiency of deep SNNs. In addition, we demonstrated that our method is compatible with a wide range of datasets, models, and spike encoding approaches, potentially making it well-suited for broader applications in energy-efficient AI using deep SNNs.

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
