# A  APPENDIX

## A.1  THRESHOLD DEPENDENT BATCH NORMALIZATION (TDBN)

tdBN is a batch normalization method designed specifically for SNNs (Zheng et al., 2021) and it can be expressed as follows:

$$x_c[t] = W \otimes s_c + B, \tag{10}$$

$$\hat{x}_c = \frac{\xi \, V_{th}(x_c - E[x_c])}{\sqrt{Var[x_c] + \epsilon}}, \tag{11}$$

$$y_c = \gamma_c \hat{x}_c + \beta_c, \tag{12}$$

where $x_c[t]$ represents the inputs at timestep $t$, $x_c = (x_c[1], x_c[2], \cdots, x_c[T])$, and $\hat{x}_c$ represents normalized $x_c$. $\xi$ is a weight factor, and $\epsilon$ is a small positive number. $\gamma_c$ and $\beta_c$ are the scale and shift parameters at $c$-th channel, respectively. tdBN successfully adjusts the firing rate of the following spiking neurons, considering their thresholds.

## A.2  FEATURE DIVERSITY LOSS

$$\frac{\partial L_{\text{FD}}}{\partial W} = \sum_k \left\{ \frac{\partial L_{\text{FD}}}{\partial p(x_k)} \frac{\partial p(x_k)}{\partial x_k} \sum_t \left( \frac{\partial x_k}{\partial s[t]} \frac{\partial s[t]}{\partial u[t]} \frac{\partial u[t]}{\partial W} \right) \right\} \tag{13}$$

$$\approx \sum_k -\log(p(x_k) - 1) p'(x_k) \sum_t I[t]/\tau, \tag{14}$$

where $I$ is the input, and $p(x)$ is a probability density function.

## A.3  DYNAMIC FEATURE ENCODING LOSS

Let $\chi_c = \frac{\beta_c}{(\gamma_c + \epsilon)}$ ,

$$\frac{\partial L_{\text{DFE}}}{\partial \chi_c} = \frac{\partial(\|\chi_c - \alpha\|_2)}{\partial \chi_c} = \frac{\chi_c}{\|\chi_c - \alpha\|_2}, \tag{15}$$

Thus,

$$\frac{\partial L_{\text{DFE}}}{\partial \beta_c} = \frac{\partial L_{\text{DFE}}}{\partial \chi_c} \frac{\partial \chi_c}{\partial \beta_c} = \frac{\chi_c}{\|\chi_c - \alpha\|_2} \frac{1}{(\gamma_c + \epsilon)}, \tag{16}$$

$$\frac{\partial L_{\text{DFE}}}{\partial \gamma_c} = \frac{\partial L_{\text{DFE}}}{\partial \chi_c} \frac{\partial \chi_c}{\partial \gamma_c} = -\frac{\chi_c^2}{\|\chi_c - \alpha\|_2} \frac{1}{(\gamma_c + \epsilon)}. \tag{17}$$

## A.4  EXPERIMENTAL SETTINGS

**Experimental Setup.** The input size of the model is set to 32x32 for CIFAR10/100, 224x224 for ImageNet, and 48x48 for CIFAR10-DVS. For CIFAR10/100 and CIFAR10-DVS, we trained each model for 300 epochs with SGD using a step-decay learning rate schedule (0.1 times every 100 epochs). For ImageNet, we trained each model for 90 epochs with SGD using a step-decay learning rate schedule (0.1 times every 30 epochs). The initial learning rate is 0.1 (0.01 for CIFAR10-DVS), and the optimizer includes L2 regularization with a lambda of 1e-4. For CIFAR10/100 and CIFAR10-DVS, the batch size was set to 100, while for ImageNet, it was set to 200. For data augmentation, CutMix (Yun et al., 2019) was applied to static datasets, while random crop and random flip (horizontal and vertical) were used for the neuromorphic dataset. During training, the time step for all datasets was set to four. Additionally, the initial leak constant $\tau$ and threshold $V_{\text{th}}(0)$ are set to 1/0.9 and 0.5. The experiments were conducted using an NVIDIA A6000 GPU. The training times for 300 epochs were approximately 6-7 hours for VGG16 on both CIFAR10 and CIFAR100 and about 20 hours for VGG16 on CIFAR10-DVS. For ResNet models, ResNet19 took around 23 hours on CIFAR10 and 20 hours on CIFAR100, while ResNet20 required roughly 12 hours on CIFAR10 and 11 hours on CIFAR100. For ImageNet, training ResNet18 with 2 GPUs took around 8 days, while training ResNet34 with the same number of GPUs took about 10 days.

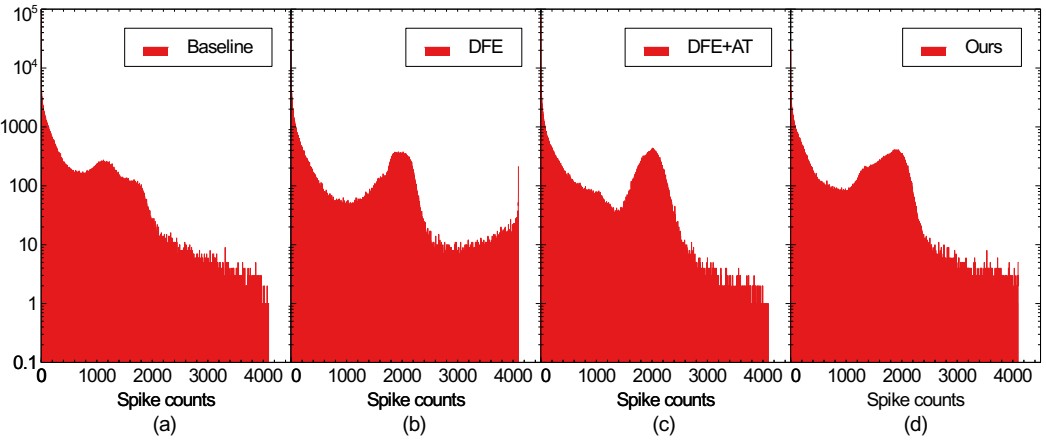

Figure 6: Distributions of each channel's spike counts on (a) baseline, (b) DFE, (c) AT+FD, and (d) ours (VGG16, CIFAR10)

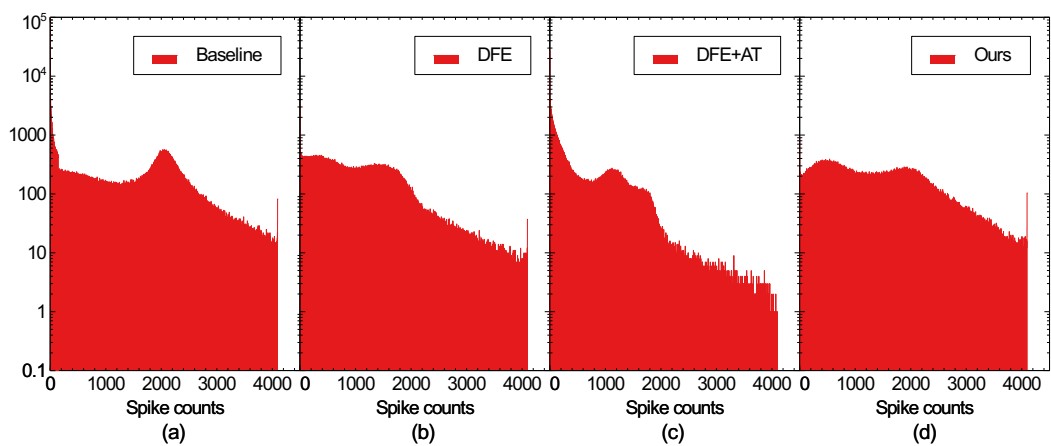

Figure 7: Distributions of each channel's spike counts on (a) baseline, (b) DFE, (c) AT+FD, and (d) ours (ResNet20, CIFAR10)

**How to set hyperparameters?** In our experiments, we used four types of hyperparameters: adjust rate $\eta$ in AT, $\lambda_{FD}$ that is the weight factor of $L_{FD}$, $\alpha$ that is a ratio of $\beta/\gamma$ in tdBN of the encoding layer, and $\lambda_{DFE}$ that is the weight factor of $L_{DFE}$. We set $\eta$ empirically by varying the value between 0.1 and 1.0 in increments of 0.1. In addition, we derived $\alpha$ from the parameters ($\beta$ and $\gamma$) of DSE channels in the baseline case. We set $\alpha$ to the average value of $\beta/\gamma$ in the DSE channels. The weight factors of total loss ($\lambda_{CE}$, $\lambda_{FD}$, and $\lambda_{DFE}$) also were determined empirically. The values of hyperparameters we used in this work are shown in Tab. 6.

Table 6: Hyperparameters in experiments

| Datasets | Architectures | $\eta$ (Eq. 6) | $\alpha$ (Eq. 4) | $\lambda_{CE}$ (Eq. 9) | $\lambda_{FD}$ (Eq. 9) | $\lambda_{DFE}$ (Eq. 9) |
|---|---|---|---|---|---|---|
| CIFAR10 | VGG16 | 0.8 | -1.0 | 1.0 | 5E-6 | 1E-4 |
| | ResNet19 | 0.8 | -0.3 | 1.0 | 3E-3 | 1E-4 |
| | ResNet20 | 0.2 | -0.4 | 1.0 | 3E-3 | 1E-3 |
| CIFAR100 | VGG16 | 0.8 | -0.8 | 1.0 | 5E-6 | 1E-4 |
| | ResNet19 | 0.8 | -0.4 | 1.0 | 3E-3 | 1E-4 |
| | ResNet20 | 0.8 | -0.3 | 1.0 | 3E-3 | 1E-4 |
| ImageNet | ResNet18 | 0.8 | -1.0 | 1.0 | 3E-3 | 1E-3 |
| | ResNet34 | 0.8 | -0.8 | 1.0 | 3E-3 | 1E-4 |
| CIFAR10-DVS | VGG16 | 0.8 | -1.0 | 1.0 | 5E-5 | 1E-2 |

### A.5 PROPORTIONS OF THE ENCODING CHANNELS ON CIFAR10-DVS

Tab. 7 compares accuracy, the number of all spike counts, the number of encoded spike counts, and the proportion of encoding channels between the baseline and ours when experimenting with

VGG16 on CIFAR10-DVS. In the baseline, all channels are encoded as PE, which is due to the discrete characteristic of event data. In contrast, in the model with H-Direct applied, UFE and DSE are present. The presence of these two encodings significantly reduces the number of spikes.

Table 7: Comparisons of accuracy, the number of all spike counts, the number of encoded spike counts, and proportions of the encoding channels on VGG16 architecture between baseline and ours. Data: CIFAR10-DVS, Baseline: STBP-tdBN.

| Methods | Accuracy | # of all spike counts | # of encoded spike counts | Proportions (in %) | | | |
|---|---|---|---|---|---|---|---|
| | | | | OFE | UFE | DSE | PE |
| baseline | 75.10%±0.08 | 413K±1K | 146K±0.4K | 0 | 0 | 0 | 100 |
| ours | 76.15%±0.31 | 273K±2K | 18K±0.1K | 0 | 2.0 | 23.8 | 74.2 |

### A.6 Spike Counts Distributions of Each Channel

Fig. 6 and Fig. 7 show the distribution of each channel's spike counts during inference on CIFAR10 using VGG16 and ResNet20 architectures, respectively. Each histogram utilizes a logarithmic scale on the y-axis to elucidate data trends. First, in Fig. 6-(b), compared to Fig. 6-(a), there is an overall reduction in spike counts of channels, but the number of channels with spike counts greater than about 2K increases. For samples that do not require many fired channels, neurons of DSE channels exhibit reduced firing. This results in a decrease in the number of channels with spike counts less than about 2K. Conversely, in cases requiring more features, the neurons of DSE channels tend to fire, which leads to the increment of channels with spike counts above about 2K. In Fig. 6-(c), the application of AT leads to an increase in the number of spikes below 2K. Particularly, AT promotes the firing of neurons in channels that have no spike counts, mainly increasing spike counts below 1K. In Fig. 6-(d), the application of H-Direct promotes the firing of low spike counts while maintaining homeostasis, resulting in a more even distribution of spike counts and enhancing the diversity of features. The results in Fig. 7 show similar trends to those in Fig. 6. In conclusion, the most efficient encoding is achieved when H-Direct is applied, leading to fewer spikes and higher performance.

### A.7 Cross Correlation

We measured the correlation between features as in (Jin et al., 2020). The cross-correlation is defined as follows:

$$\rho(s_c) = \frac{1}{N_b N_c} \sum_{c=0}^{N_c} \sum_{b=0}^{N_b} \frac{|s_{c,b}'^T s_{c,b}'|}{\|s_{c,b}'\|_2 \|s_{c,b}'\|_2}, \qquad (18)$$

where $\mathbf{s}_c \in \mathbb{R}^{f \times f \times N_c}$ represents the features in the $c$-th channel of the encoding layer. $s_c'$ denotes the reshaped features of $s_c$ into $\mathbf{s}_c' \in \mathbb{R}^{f^2 \times N_c}$. $N_b$ represents the $b$-th sample in the batch and $s_{c,b}'$ denotes $s_c'$ for the $b$-th sample. A smaller cross-correlation value indicates fewer redundant features, suggesting more efficient encoding.

### A.8 Feature Maps of Encoded Spikes

Fig. 8 and Fig. 9 show the encoded feature maps (a) and the proportion of each categorized encoding (b) from ablation studies using VGG16 and ResNet20 architectures on CIFAR10 dataset, respectively. In the case of VGG16 (Fig. 8), OFE, UFE, DSE, and PE appear in the baseline, whereas in the model to which DFE is applied, OFE and UFE disappear and the proportion of DSE increases about 16.8% compared to the baseline. In addition, in the model with DFE and AT, it can be seen that DSE decreases by about 3.13%. In Ours, OFE and UFE disappear, and DSE accounts for the largest portion among all types (about 62.9%). In the case of ResNet20 (Fig. 9), OFE, DSE, and PE appear in the baseline. Although in the baseline feature map of Fig. 9-(a), it might appear that UFE persists at 2nd row and 2nd column, this observation holds true only when comparing specific samples (sample1 and sample2). As can be seen in Fig. 9-(b), when the entire dataset is considered, there is no presence of UFE, indicating that the channel that appears to be UFE is actually being

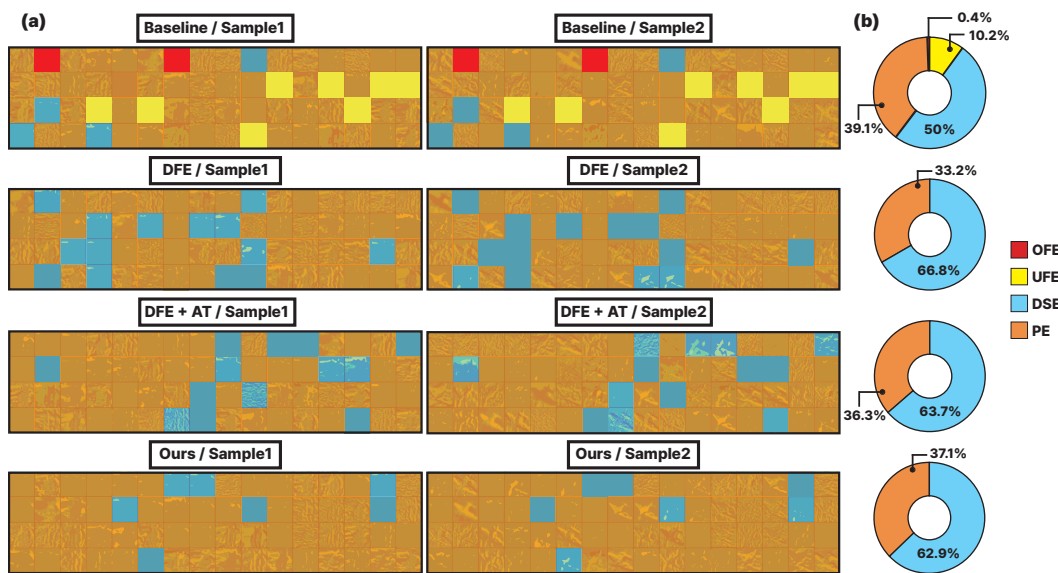

Figure 8: (a) Examples of encoded feature maps and (b) the average proportion of each categorized encoding in ablation studies (VGG16, CIFAR10).

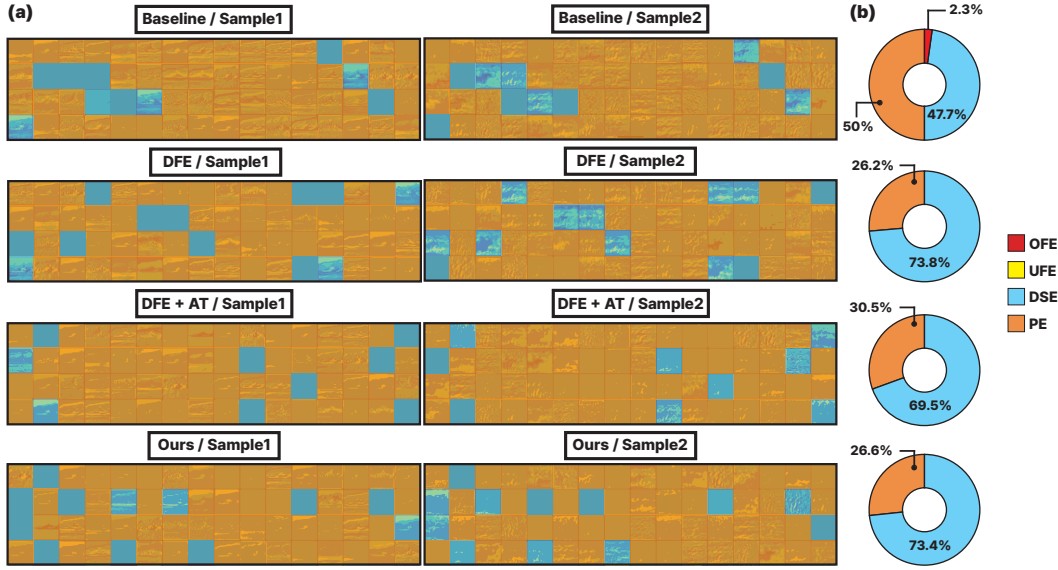

Figure 9: (a) Examples of encoded feature maps and (b) the average proportion of each categorized encoding in ablation studies (ResNet20, CIFAR10).

encoded in different samples. When DFE is applied, UFE disappears, and the proportion of DSE increases by about 23.8%. In addition, PE increases by about 4.3% than DFE in the model with DFE and AT. In ours, UFE disappears, and DSE accounts for the largest portion (about 51.6%) of all types. The result tends to be similar to VGG16, which indicates the effectiveness and compatibility of our methods across model architectures.