# OpenReview forum: "H-Direct: Homeostasis-aware Direct Spike Encoding for Deep Spiking Neural Networks"
_ICLR.cc/2025/Conference — Submitted to ICLR 2025_

### Official Review · Reviewer_N9Cn · 2024-10-16

**Soundness:** 2
**Presentation:** 1
**Contribution:** 1
**Rating:** 3
**Confidence:** 5

**Summary:**

The paper proposes H-Direct, a homeostasis-aware spike encoding technique for deep Spiking Neural Networks (SNNs). It incorporates mechanisms like dynamic feature encoding loss, adaptive thresholding, and feature diversity loss to improve the performance and efficiency of deep SNNs. While the method shows potential in improving spike encoding performance, there are several concerns that prevent this work from being accepted at this stage.

**Strengths:**

The concept of integrating homeostasis to regulate spike encoding is interesting and innovative. The proposed approach seems to achieve meaningful improvements in terms of classification accuracy and energy efficiency in the experiments conducted.

**Weaknesses:**

While the introduction of homeostasis in spike encoding is intriguing, the overall novelty of the approach is limited. Many of the techniques applied, such as adaptive thresholds and feature diversity loss, have been explored in various forms in prior works on SNNs and neuromorphic computing. There is insufficient discussion on how this method truly differentiates itself from previous work like direct encoding approaches and surrogate gradients. The proposed method lacks a strong theoretical foundation. Although the paper discusses homeostasis from a biological perspective, the paper does not thoroughly justify the mathematical and theoretical basis for why introducing homeostasis in the encoding stage would lead to significant improvements in SNN performance. A deeper explanation of the relationship between the proposed encoding mechanisms and their impact on network learning dynamics is needed.

The experiments focus heavily on classification tasks using typical SNN benchmarks (CIFAR-10, CIFAR-100, etc.), which limits the generalization of the proposed approach. There are no experiments on more challenging neuromorphic datasets or tasks beyond classification, which could better validate the robustness and applicability of H-Direct. The performance gains, while present, are relatively marginal compared to existing state-of-the-art methods, especially when considering the complexity introduced by the additional loss functions and adaptive mechanisms. The paper’s presentation could be significantly improved. The descriptions of the core methods (like dynamic feature encoding loss and adaptive thresholds) are somewhat unclear and hard to follow without repeated reading. Additionally, the figures do not always effectively support the key points being made and could benefit from clearer labeling and explanations. The related work section does not adequately cover recent advances in energy-efficient SNN training or alternative spike encoding methods. Key works on the use of advanced encoding strategies and their implications on energy efficiency should be cited and discussed.

**Questions:**

See weaknesses.

**Details Of Ethics Concerns:**

Although the paper explores an important topic, the lack of clear novelty, limited theoretical depth, and relatively small performance improvements over existing techniques make this work insufficient for acceptance at ICLR. Further refinement in the clarity of exposition, a stronger theoretical foundation, and more comprehensive experimental validation would be necessary for future consideration.

---

> ### Author Response · Authors · 2024-11-18
>
> Thank you for your constructive feedback on our manuscript. We have carefully reviewed your comments and would like to ask for clarification on several points.
>
> 1. *“Many of the techniques~ neuromorphic computing.”*
> To the best of our knowledge, we have not encountered cases where adaptive thresholds or feature diversity loss were applied specifically to the encoding layer. Existing studies have not addressed issues related to direct encoding, which has been widely used in input spike encoding for deep SNNs. In our work, we believe this work is the first attempt to identify the limitations of conventional direct encoding and propose methods to address them. If there are relevant papers discussing adaptive thresholds or feature diversity loss applied to encoding layers, please let us know.
> 2. *“The experiments focus heavily~ applicability of H-Direct.”*
> The datasets used in this study are benchmarks commonly used to validate performance in deep SNNs, as referenced in [Rathi & Roy, 2021; zheng et al.,2021; Feng et al., 2022; Guo et al.,2022a; Guo et al., 2023; Zhou et al., 2023]. If you are aware of benchmarks that could better demonstrate the generalization performance of our proposed method, we would greatly appreciate your suggestions. Additionally, CIFAR10-DVS is, to our understanding, a neuromorphic dataset with considerable room for performance improvement. While automotive detection tasks such as Gen4 exist as an event-based vision data, these tasks beyond the scope of this work.
> 3. *“The performance gains~ adaptive mechanisms.”*
> Our experiments were conducted with baselines achieving SOTA-level performance. Therefore, as you noted, the improvements may appear marginal. However, we note that it is similarly challenging for other SOTA papers to demonstrate significant gains over competitive baselines [Guo et al.,2022a; Guo et al., 2023]. As shown in Table 2 of our manuscript, despite only improving the spike input encoding, our method enhances performance while reducing the number of spikes, thus improving energy efficiency. Furthermore, Table 2 also demonstrates that our method can be universally applied to approaches employing direct training, achieving both performance and efficiency improvements. We anticipate that our approach can enhance future SOTA methods.
> 4. *“The paper’s presentation~ explanations.”*
> If you can clarify which parts need improvement, we will consider revising the manuscript.
>
> We believe your responses to the above questions will be instrumental in enhancing our manuscript. Thank you for taking the time to consider our queries, and we look forward to your feedback.

---

> > ### Author Response · Authors · 2024-11-22
> >
> > We provided detailed responses to your concerns and requested further discussion, but as we have not yet received a reply.
> > As mentioned by Reviewer MhWu, we have clearly demonstrated improvement in the efficiency and stability of our proposed method through extensive experiments and thorough validation. Furthermore, as pointed out by Reviewer KtLe, our proposed method has demonstrated its generalizability by improving performance and efficiency across various datasets, models, and training algorithms.
> > If your concerns remain despite these comments from other reviewers, please respond to the questions raised in our previous response.

---

### Official Review · Reviewer_KtLe · 2024-10-17

**Soundness:** 2
**Presentation:** 2
**Contribution:** 2
**Rating:** 6
**Confidence:** 5

**Summary:**

This study addresses the limitations of existing direct spike encoding by first analyzing traditional direct spike encoding methods and categorizing the encoded spike channels into four types: over-activated, under-activated, dynamically selective, and persistent encoding. Subsequently, the concept of homeostasis was introduced into direct spike encoding, and this paper proposes a homeostasis-aware direct spike encoding method called H-Direct. This method achieves stable and appropriate encoding by suppressing over- and under-activation while encouraging dynamic feature selection.

**Strengths:**

1. Experiments show that the proposed steady-state mechanism improves the training performance and efficiency of deep SNNs on various datasets, models, and training algorithms, demonstrating the effectiveness and universality of the method.
2. A coding method inspired by the human brain is proposed.

**Weaknesses:**

1. This paper significantly reduces the number of spike data, but lacks theoretical power consumption calculation. Can it give specific power consumption results and compare them with some advanced works [1]?
2. The experimental effect of this paper is limited, and it is not compared with some advanced works [2][3] on many datasets.
I hope the author can introduce why H-direct works from a theoretical perspective.

[1] Yao M, Hu J, Zhou Z, et al. Spike-driven transformer[C]. Advances in neural information processing systems, 2024, 36.
[2] He H. Adaptive Spiking Neural Networks with Hybrid Coding[J]. arXiv preprint arXiv:2408.12407, 2024.
[3] Hu J K, Yao M, Qiu X, et al. High-Performance Temporal Reversible Spiking Neural Networks with $\mathcal {O}(L) $ Training Memory and $\mathcal {O}(1) $ Inference Cost[C]. Forty-first International Conference on Machine Learning.

**Questions:**

see weakness below.

---

> ### Author Response · Authors · 2024-11-22
>
> Thank you for the constructive feedback on our manuscript.
>
> **Q1. This paper significantly reduces the number of spike data, but lacks theoretical power consumption calculation. Can it give specific power consumption results and compare them with some advanced works [1]?**
>
> A1. Following the calculation method mentioned in [1], we attach the power consumption results and comparisons as below. The comparison targets are studies that used the same architecture and reported power consumption results. We assume that the data for various operations are based on a 32-bit floating point implementation in 45nm technology, where E_MAC = 4.6 pJ and E_AC = 0.9 pJ as [1]. Experimental results show that our method achieves higher performance with lower energy consumption compared to previous studies, and we will reflect this in the manuscript.
>
> Table R1. Comparison with previous studies in terms of power consumption and accuracy based on the ResNet34 architecture.
> | Dataset | Architecture | Methods | Timesteps | Energy (mJ)| Acc. (%) |
> |-|-|-|-|-|-|
> |ImageNet|ResNet34|tdBN|6|6.39|63.72|
> |||TEBN|4|7.05|64.29|
> |||MPBN|4|6.56|64.71|
> |||H-direct (ours)|4|2.45|68.43|
>
> **Q2. The experimental effect of this paper is limited, and it is not compared with some advanced works [2][3] on many datasets. I hope the author can introduce why H-direct works from a theoretical perspective.**
>
> A2. Our experiments were conducted using a baseline with state-of-the-art (SOTA) performance. Thus, as you mentioned, the performance improvement may not appear substantial, but other SOTA studies also do not show significant performance improvements compared to their baselines [Guo et al., 2022a; Guo et al., 2023]. Furthermore, as shown in Table 2 of the manuscript, despite only improving spike input encoding, our method achieved improved performance with fewer spikes compared to the baseline, thereby enhancing energy efficiency and reducing computational costs. Additionally, as also shown in Table 2, our method can be generally applied to methods that use direct training, achieving both performance and efficiency improvements. This approach will also be applied to other methods presented in the future.
>
> We also thoroughly reviewed and compared with the papers you referenced for comparison [2], [3]. First, [2] proposes a hybrid coding method that combines ASNN, which includes learnable components with time-to-first-spike (TTFS) encoding and direct encoding. Implementing this method requires additional computations for TTFS coding (encoding and decoding). Moreover, hybrid coding involves using both encoding methods, which we believe leads to considerable computational overhead. Since this study does not provide spike counts or energy consumption details, a quantitative comparison is limited. In contrast, our method can be applied without modifying the deep SNN model structure or requiring additional overhead during inference, which is a significant advantage. The significance of our approach is that it leverages the model capacity of deep SNNs to improve encoding efficiency and performance, thereby enhancing the overall performance and efficiency of deep SNNs.
>
> Next, [3] proposes a method to improve the training efficiency of deep SNNs based on a ConvNeXt-style [R1]. The proposed model has structural characteristics different from the commonly used ResNet and also deviates from the model proposed in [R1] by excluding Layer Normalization, among other changes. Such deep SNN model structures have not yet been widely adopted in related research. Despite the specificity of the model, this study also uses direct encoding to achieve high performance. Therefore, we believe that our methodology can be applied to the method and model structure. Even if BN is not used, the DFE methodology of regulating spike firing by adjusting the input distribution of neurons can still be applied, and in such cases, it could be implemented using a loss function such as KL-divergence. Our approach to enhancing the performance and efficiency of deep SNNs through improved spike encoding can be applied to various models, datasets, and algorithms that use direct encoding.
>
> Our study represents early research of efficient direct encoding, and through comprehensive experiments and analysis, we have thoroughly demonstrated that performance and efficiency can be improved solely by enhancing direct encoding. We will reflect the impact of H-Direct from a theoretical perspective in future research.
>
> Thank you for taking the time to review our work.
>
> [R1] Liu, Zhuang, et al. "A convnet for the 2020s." *Proceedings of the IEEE/CVF conference on computer vision and pattern recognition*. 2022.

---

> > ### Comment · Reviewer_KtLe · 2024-11-25
> >
> > I also found this paper very similar to Homeostasis-aware Direct Spike Encoding for Deep Spiking Neural Networks https://openreview.net/pdf?id=Uvsa7vFWcT? They have a lot of similarities in both images and content. For this reason, I would consider lowering my score

---

> ### Author Response · Authors · 2024-11-25
>
> The paper the reviewer mentioned is our work that will be presented at the NeurIPS 2024 workshop (NeuroAI).
>
> As stated in the call for papers for that workshop, papers presented at this workshop are ***not considered for publication (”non-archival”)***.
>
> Call for papers for the workshop (https://neuroai-workshop.github.io/call-for-papers/)
>
> “… We invite researchers in the field of NeuroAI to submit their papers to this workshop. The workshop papers are ***non-archival*** and authors are also encouraged to submit short versions of their NeurIPS main conference submissions concurrently to this workshop. …”
>
> Also, according to ICLR’s dual submission policy, ***presented papers at workshops without publication proceedings do not violate the submission policy***.
>
> Dual Submission Policy of ICLR (https://iclr.cc/Conferences/2025/CallForPapers)
>
> “… However, papers that cite previous related work by the authors and papers that have appeared on non-peer reviewed websites (like arXiv) or ***that have been presented at workshops (i.e., venues that do not have publication proceedings) do not violate the policy***. … ”
>
> Therefore, there is no problem with our submission of this manuscript, and we ask that the reviewers appropriately evaluate this paper with our response above.

---

> > ### Comment · Reviewer_KtLe · 2024-11-25
> >
> > Thanks for the explanation. I’ve adjusted my score back to 5, but it won’t go higher because the performance improvement of the method proposed in this paper is too limited. If it can be demonstrated on hardware, it may serve as a strong motivation.

---

> > > ### Author Response · Authors · 2024-11-25
> > >
> > > As mentioned earlier, our evaluation was performed according to SOTA-level criteria, so our improvement is meaningful compared to other related papers, as mentioned in our response. Additionally, the significance of our method lies in its compatibility with various methods, architectures, and datasets, with minimal overhead. As reviewer ‘MhWu’ pointed out, we have thoroughly validated this through comprehensive experiments. Based on these strengths, we expect our method to become a novel approach for improving the performance and efficiency of models that use direct encoding. We kindly ask you to reconsider your evaluation of our method, taking these points into account.

---

> > > > ### Comment · Reviewer_KtLe · 2024-11-25
> > > >
> > > > Most of my concerns have been addressed, so I'm going to raise my score, but the results are not compared with the latest Sota works [1] [2]. Please experiment with state-of-the-art open-source work such as GAC [1] and PFA [2], which achieve nearly 80% accuracy on the CIFAR100 dataset.
> > > >
> > > > [1] Gated attention coding for training high-performance and efficient spiking neural networks. AAAI 2024.
> > > >
> > > > [2] Tensor decomposition based attention module for spiking neural networks. Knowledge-Based Systems 2024.

---

> > > > > ### Author Response · Authors · 2024-11-28
> > > > >
> > > > > We appreciate your decision and the opportunity to respond. When comparing the performance of the two mentioned papers on CIFAR100 with our method, we observe a slight performance difference. However, our approach not only improves performance but also reduces the number of spikes, thereby enhancing efficiency. This has been thoroughly validated through extensive experiments.
> > > > >
> > > > > The spike counts for the two mentioned papers were not reported, making direct numerical comparisons impossible. However, given the additional attention layers required in methods like GAC and PFA, it is reasonable to assume that their spike counts are relatively high. Moreover, both methods focus on improving encoding by introducing new layers.
> > > > >
> > > > > In contrast, our method does not require additional layers and can be directly applied to the existing models, offering greater generalizability. Therefore, we believe that our approach could also be applied to the methods you mentioned, potentially yielding further performance improvements. This broad applicability highlights the significance of our research, as it paves the way for generalizable enhancements across various studies.

---

### Official Review · Reviewer_ey4J · 2024-11-01

**Soundness:** 2
**Presentation:** 3
**Contribution:** 2
**Rating:** 3
**Confidence:** 5

**Summary:**

This paper presents an analysis of direct encoding and introduces a concept of homeostasis to direct spike encoding, including feature encoding loss, adaptive threshold, and feature diversity loss. The experimental results demonstrate that the proposed encoding improves performance and efficiency, but performance improvements are limited.

**Strengths:**

The direct encoding issue of SNN that the article focuses on is valuable, and the experiments are also sufficient.

**Weaknesses:**

1. Many statements in the paper lack sufficient persuasiveness,  such as those described in the abstract and introduction.
- 'Despite the importance of the encoding, efficient encoding methods HAVE NOT BEEN studied.'
As far as I know, the time-to-first-spike (TTFS) method that utilizes the first spike time to encode information is regarded as a highly energy-efficient encoding method.
- 'Thus, the energy problem has become the MOST urgent issue to be addressed for sustainable development and utilization of AI in our lives.'
Aside from the energy problem, the sustainable development of AI also faces multiple challenges, such as data security, interpretability, model generalization, and regulatory and formal laws.
- 'Direct encoding learns encoding methods from data, which leads to superior performance over other encoding approaches.'
Can the author elaborate on this?

2. The proposed method lacks sufficient innovation, since the approach of setting dynamic neuronal thresholds and adjusting loss functions to enhance performance is widely used.

3. The experimental results do not demonstrate a substantial improvement in accuracy, and as model size and dataset scale increase, this improvement diminishes further. This raises concerns about the practical utility of the proposed method.

4. Does the performance improvement depend on a large number of training epochs? Could the authors provide more experimental details? Additionally, does the introduction of the proposed loss function add computational overhead?

**Questions:**

In addition to the problem mentioned in weakness, I have the following questions:
1. In Figure 1, in the image to the left of (b), the red box denotes OFE (Over-fired Encoding). Why is the RED box filled with ORANGE blocks (Orange represents normal spike counts)?

2. The criteria for the divisions are not clearly articulated. For instance, what spike count corresponds to DSE? Furthermore, when the spike count is zero, how to distinguish between OFE and DSE?

3. Given that the performance improvements demonstrated in the experimental results are not substantial, the motivation behind this work raises questions. Intuitively, not all spiking neurons contribute to the output; some neurons at specific pixels may not respond or fire, which is also a reflection of the “sparseness” advantage of SNN. Therefore, is it necessary to activate all neurons associated with OFE and UFE?

---

> ### Author Response · Authors · 2024-11-22
>
> Thank you for the constructive feedback.
>
> **W1. Many statements in the paper lack sufficient persuasiveness, such as those described in the abstract and introduction.**
>
> A1. It seems our wording may have led to some misunderstanding. First, regarding *“Despite the importance of encoding, efficient encoding methods HAVE NOT BEEN studied,”* as you mentioned, temporal coding methods like TTFS have indeed been studied. We intended to highlight that efficient encoding methods specifically for direct encoding, which is widely used in deep SNNs, have not yet been thoroughly explored.
> Additionally, the statement *“Thus, the energy problem has become the MOST urgent issue to be addressed for the sustainable development and utilization of AI in our lives”* was meant to convey that the energy issue is one of the major challenges in AI. We apologize for the confusion caused by our wording and will revise the manuscript to make our intention clearer.
> Regarding *“Direct encoding learns encoding methods from data, which leads to superior performance over other encoding approaches,”* we can confirm through Table 3 of [Qiu et al., 2024] that direct encoding outperforms other encoding schemes. Moreover, almost all recent studies demonstrating state-of-the-art performance use direct encoding [Rathi & Roy, 2021; Zheng et al., 2021; Feng et al., 2022; Guo et al., 2022a; Guo et al., 2023; Zhou et al., 2023].
>
> **W2. The proposed method lacks sufficient innovation, since the approach of setting dynamic neuronal thresholds and adjusting loss functions to enhance performance is widely used.**
>
> A2. We have not encountered any previous studies where adaptive thresholds and feature diversity loss have been applied to the encoding layer. Existing research has not addressed the widely used input spike encoding method, direct encoding, in current deep SNNs. To the best of our knowledge, this work is the first attempt to identify issues in conventional direct encoding and propose methods to address them. If there are related papers on applying adaptive thresholds or feature diversity loss to the encoding layer that we are unaware of, please let us know.
>
> **W3. The experimental results do not demonstrate a substantial improvement in accuracy, and as model size and dataset scale increase, this improvement diminishes further. This raises concerns about the practical utility of the proposed method.**
>
> A3. Our experiments were conducted with a baseline of state-of-the-art (SOTA) performance. Thus, as you mentioned, the performance improvement may not appear substantial, but it is also difficult to see significant improvements in other related papers compared to their baselines [Guo et al., 2022a; Guo et al., 2023]. Moreover, as shown in Table 2 of the manuscript, despite improving only the spike input encoding, our method achieved better performance than the baseline with fewer spikes, thereby increasing energy efficiency and reducing computational costs. As can be seen from the same table, our method can be applied universally to methods using direct training, achieving both performance and efficiency improvements. This approach can also be applied to other SOTA methods presented in the future. Additionally, as shown in Table 2, the improvement effect does not always decrease as the model size and dataset scale increase. For instance, in the case of VGG16, the performance improvement is greater for CIFAR100 compared to CIFAR10, and on CIFAR100, the performance improvement for ResNet20 is greater than for ResNet19.

---

> ### Author Response · Authors · 2024-11-22
>
> **W4. Does the performance improvement depend on a large number of training epochs? Could the authors provide more experimental details? Additionally, does the introduction of the proposed loss function add computational overhead?**
>
> A4. The configuration of our experiments was set similarly to those commonly used in related studies, and training was conducted until the training loss sufficiently converged [Deng et al., 2022; Guo et al., 2023]. In addition, the experimental results for tdBN, IM-loss, and RMP-loss provided in Table 2 were all conducted under the same settings and were fairly evaluated. Under the same experimental conditions, applying our method showed improvements in performance. More detailed experimental setups are provided in Appendix A.4. If you have any further questions about specific details beyond the provided experimental information, please let us know, and we will be happy to answer them.
> Our approach only modifies the encoding layers. Let $N_e$ be the number of encoding neurons. Then, the computational overhead of AT and FD is O($N_e$). The overhead of DFE is O($C_e$), where $C_e$ indicates the number of channels in the encoding layer. Considering that training demands O(W) computations and W is generally larger than $N_e$ and $C_e$, the overhead of our method is insignificant. We also demonstrated the overhead using wall clock time during training. As you can see in Table R1, when our method is applied, training time increases by only 2.26%. According to our measurements, most of the overhead was caused by FD, likely due to more complex operations, such as fitting the probabilistic density function. Detailed results are shown in the table below.
>
> Table R1: Ablation study for overhead on CIFAR10
> | Dataset | Architecture | Methods  | ms/step | $\Delta$ (↑)| $\Delta$ (%)|
> |-|-|-|-|-|-|
> |CIFAR10|VGG16|Baseline|194.8|-|-|
> |||w/ DFE|196.3|+1.5|+0.77|
> |||w/ FD|196.9|+2.1|+1.08|
> |||w/ DFE+FD|198.8|+4.0|+2.05|
> |||w/ AT+FD+DFE (ours)|199.2|+4.4|+2.26|
>
> **Q1. In Figure 1, in the image to the left of (b), the red box denotes OFE (Over-fired Encoding). Why is the RED box filled with ORANGE blocks (Orange represents normal spike counts)?**
>
> A5. The colors of each block in Figure 1-(a) represent the spike count (refer to the spike count legend in the upper right of Figure 1). In the example of Figure 1-(a), the “RED box filled with ORANGE blocks (actually YELLOW blocks)” represents neurons with a spike count of four (refer to the legend in the upper right of Figure 1). In other words, the box (channel) marked in red represents OFE. We apologize for the confusion caused by the figure. We will update the related figure to more clearly explain our method.
>
> **Q2. The criteria for the divisions are not clearly articulated. For instance, what spike count corresponds to DSE? Furthermore, when the spike count is zero, how to distinguish between OFE and DSE?**
>
> A6. Based on our findings in direct encoding, we categorized encoding channels into four types (PE, DSE, UFE, and OFE) mainly depending on the encoding rate of each channel. Encoding rate $R_c^d$ is defined as follows:
>
> $$
> R_c^d =\sum_{d} H(S_{c}^{d}) / |D|.
> $$
>
> $$
> S^d_c=\sum_{i\in Channels_c}\sum_{t}^{T}s_{i}^{d}[t],
> $$
>
> $$
> H(x) =  \begin{cases} 0 & x\leq 0 \\\\ 1 & x>0 \end{cases}.
> $$
> - UFE: $R_c^d$ == 0, meaning they cannot encode any spikes for all input data. This kind of channel prohibits fully utilizing of given model capacity.
> - DSE: 0 < $R_c^d$ < 1, meaning they decide whether to encode or not depending on the input. To adjust the encoding rate appropriately, we let them learn whether to encode or not through training data.
> - PE: $R_c^d$ == 1, meaning they always encode spikes for all input data.
> - OFE: $R_c^d$ == 1 and *standard deviation* ($\sigma$) of $\sum_t^T {s_i^d[t]}$ is 0,  meaning that all neurons in Channel *c* fire with the same spike value(e.g., 1,2,…,t) for all input data, resulting in the encoding as meaningless features.
>
> As explained, channel categorization is not a concept that can be defined for a single input. We hope that our additional explanation has been helpful for your understanding.

---

> > ### Author Response · Authors · 2024-11-22
> >
> > **Q3. Given that the performance improvements demonstrated in the experimental results are not substantial, the motivation behind this work raises questions. Intuitively, not all spiking neurons contribute to the output; some neurons at specific pixels may not respond or fire, which is also a reflection of the “sparseness” advantage of SNN. Therefore, is it necessary to activate all neurons associated with OFE and UFE?**
> >
> > A7. We fully agree with the reviewer's opinion. We aim to encode only the pixels that are necessary to increase sparsity without compromising accuracy. H-Direct suppresses unnecessary firing and increases the ratio of DSE, which selectively encodes as needed, enabling efficient encoding. We let deep SNNs learn how to extract important features from input and encode them into spikes.  Additionally, encodings that hinder task performance can be eliminated during this process. Consequently, high-spike features are replaced with low-spike features (Figure 5-(b)), leading to increased sparsity and improved accuracy (Table 1). As shown in Table 1, applying our method not only reduces the number of encoded spikes but also decreases the total number of spikes. Therefore, our method increases sparseness, rather than decreasing it.
> >
> > Thank you for taking the time to review our work.

---

> > > ### Comment · Reviewer_ey4J · 2024-11-25
> > >
> > > I sincerely appreciate the authors' response. But I still prefer to maintain the original score due to two main concerns: lack of innovation and marginal performance improvement.
> > >
> > > Innovation:
> > >
> > > 1. ADAPTIVE THRESHOLD IN ENCODING NEURONS. [1] introduce the dynamic firing threshold into TTFS encoding to solve the training problem of TTFS-based deep SNN. While the H-Direct approach differs from the method presented in this paper, it appears that the authors merely integrate an adaptive firing threshold with direct encoding, subsequently applying it to the encoding layer. More specifically, the proposed adaptive threshold seems not to fully exploit the unique characteristics of direct encoding. This approach could be extended to other encoding strategies, such as temporal encoding, population encoding, and others. Why didn't the author design the adaptive threshold based on the characteristics of direct encoding? Perhaps this makes more sense.
> > > 2. FEATURE DIVERSITY LOSS. This loss function is the sum of multiple losses and is not well designed. Also, similar methods have been widely studied. It seems that the author just combined it with his own work [2,3,4].
> > >
> > > > [1] Temporal-coded spiking neural networks with dynamic firing threshold: Learning with event-driven backpropagation. ICCV2023.
> > > >
> > > > [2] Recdis-snn: Rectifying membrane potential distribution for directly training spiking neural networks. CVPR 2022.
> > > >
> > > > [3] Rmp-loss: Regularizing membrane potential distribution for spiking neural networks. ICCV 2023.
> > > >
> > > > [4] IM-loss: information maximization loss for spiking neural networks. NeurIPS 2022.
> > >
> > >
> > >
> > > Performance:
> > >
> > > The method proposed in this paper has limited improvement in performance, which makes people worry about the effectiveness of the method. Of course, the authors claim that the method is based on the SOTA-level criteria, so the performance improvement is limited. However, it is important to note that this work is based on the commonly used direct encoding. The proposed complex method and multi-loss fail to deliver performance gains, raising questions about its practical value. In particular, the constraints imposed by the FEATURE DIVERSITY LOSS on training convergence further undermine the applicability of the proposed method to complex structures and tasks.

---

> > > > ### Author Response · Authors · 2024-11-28
> > > >
> > > > The adaptive threshold mechanism is widely used in various neural network models [R1,R2]. In this study, we apply it to improve the spike encoding.
> > > >
> > > > Through DFE, the proportion of DSE increases, leading to improved model efficiency. However, this process does not ensure feature diversity, which may result in performance degradation. To address this issue, we introduce an adaptive threshold mechanism that dynamically adjusts the threshold for each channel, thereby encouraging diverse feature activation. To address this issue, we introduce an adaptive threshold that dynamically adjusts the threshold on a per-channel basis. Additionally, through asymmetric adjustment, it promotes firing in under-activated channels, maintaining neuron activity and preventing performance degradation. Thus, the proposed adaptive threshold is specifically designed to address issues inherent to direct encoding, reflecting an approach tailored to its characteristics.
> > > >
> > > > When compared to the Dynamic Firing Threshold (DFT) proposed in [1], the two techniques differ significantly in their purpose and mechanism. DFT focuses on improving learning efficiency by linearly decreasing the firing threshold of neurons over time during training, thereby gradually increasing the number of active neurons in the network. It removes the need for a complex causal set-finding process and adjusts the firing of neurons in each layer within unique time windows to ensure all presynaptic neurons contribute to postsynaptic neurons. In contrast, our proposed adaptive threshold specifically addresses challenges in the direct encoding process. To the best of our knowledge, this is the first study to apply such an approach to direct encoding, which we believe is a meaningful contribution.
> > > >
> > > > First, we would like to clarify that the proposed FD Loss is not a simple combination of existing loss functions but a novel loss function introduced by our study. In H-direct, we propose two loss functions: DFE Loss and FD Loss. Regarding your comment that similar methods have been extensively studied, we agree that the design of loss functions is a critical research topic for improving learning performance. Indeed, many prior studies have designed diverse loss functions to effectively enhance training outcomes [Guo et al.,2022a; Guo et al., 2023].
> > > >
> > > > Our FD Loss is specifically designed to encourage the encoding of diverse features during the encoding process. It promotes neurons to encode a broader range of features, thereby enhancing the representational capacity of the model. To the best of our knowledge, this is the first study to propose a loss function aimed at increasing feature diversity during encoding. Through this, our method effectively encodes diverse features, leading to performance improvements.
> > > >
> > > > Additionally, regarding your comment that our FD Loss is “not well designed,” we would like to seek clarification. As mentioned earlier, our FD Loss was designed to enhance the diversity of encoded features, and we have experimentally confirmed that it improves performance. If you could provide further details on the aspects in which you find the FD Loss to be “not well designed,” it would greatly help us refine and improve our study. We deeply appreciate your valuable feedback and look forward to your detailed input.
> > > >
> > > > [R1]Shaban, Ahmed, Sai Sukruth Bezugam, and Manan Suri. "An adaptive threshold neuron for recurrent spiking neural networks with nanodevice hardware implementation." *Nature Communications* 12.1 (2021): 4234.
> > > >
> > > > [R2]Ding, Jianchuan, et al. "Biologically inspired dynamic thresholds for spiking neural networks." *Advances in Neural Information Processing Systems* 35 (2022): 6090-6103.

---

### Official Review · Reviewer_MhWu · 2024-11-03

**Soundness:** 3
**Presentation:** 3
**Contribution:** 3
**Rating:** 5
**Confidence:** 5

**Summary:**

This paper proposes a homeostasis-aware direct spike encoding method (H-Direct) for spiking neural networks (SNNs) to improve encoding efficiency and stability. The authors introduce three novel components: Dynamic Feature Encoding (DFE) Loss, Adaptive Threshold (AT), and Feature Diversity (FD) Loss. Extensive experiments across multiple datasets demonstrate the effectiveness of H-Direct in enhancing the performance of deep SNNs, and the ablation study further shows the contribution of each component.

**Strengths:**

The experimental validation is thorough, with tests conducted on multiple datasets and network architectures, providing strong evidence for the proposed method’s efficacy. The authors also conducted an ablation study that helps to highlight the importance of each component within H-Direct, particularly in enhancing encoding stability and feature diversity. The overall approach is well-motivated and provides an innovative solution to improve SNN encoding by borrowing the concept of homeostasis from biological systems.

**Weaknesses:**

While H-Direct addresses important encoding issues in the encoding layer, similar issues may also exist in deeper layers, raising the question of why these corrections are not applied throughout the network. Additionally, the ablation study in Table 4 only shows improvements on CIFAR-10 with VGG16, while ResNet20 shows decreased performance for certain modules. This inconsistency brings the role of each module into question. DFE is also heavily dependent on BN, raising concerns about its applicability to networks without BN. Further clarification on why FD Loss fits a probability density function and how H-Direct performs on architectures like RNNs or Transformers is also needed. If the authors are unable to sufficiently address these concerns, I may consider lowering my score. However, should the authors provide thorough explanations and additional experiments that address these points, I would raise my score.

**Questions:**

1.	Why are the corrective mechanisms not applied beyond the encoding layer, given that over-firing and under-firing issues may also exist in deeper layers?
2.	In Table 4, why do certain modules of H-Direct decrease performance in ResNet20 compared to the baseline? Does this suggest limitations in module effectiveness across architectures?
3.	How would DFE perform in networks without Batch Normalization?
4.	What is the purpose of fitting FD to a probability density function, and which PDF is used? What is the impact of different PDFs?
5.	Does H-Direct show improvements in other architectures, such as RNNs or Transformers?
6.	Could the authors analyze the robustness of hyperparameters for the DFE, AT, and FD losses?

---

> ### Author Response · Authors · 2024-11-22
>
> Thank you for the constructive feedback.
>
> **Q1. Why are the corrective mechanisms not applied beyond the encoding layer, given that over-firing and under-firing issues may also exist in deeper layers?**
>
> A1. Our research aims to emphasize the importance and necessity of efficient input spike encoding in deep SNNs. In this work, we discovered the problems in the conventional direct encoding and experimentally demonstrated that improving input encoding can lead to enhanced performance and efficiency. We anticipate that applying our proposed method to hidden layers will further enhance performance and efficiency. This will be investigated in our future research.
>
> **Q2. In Table 4, why do certain modules of H-Direct decrease performance in ResNet20 compared to the baseline? Does this suggest limitations in module effectiveness across architectures?**
>
> A2. There is a trade-off between the efficiency (i.e., the number of spikes) and performance (i.e., accuracy) in deep SNNs. To improve these two simultaneously, we proposed H-direct, which consists of complementary methods. The main purpose of DFE is to increase efficiency by reducing inappropriate encodings and encoding rates. In cases where the proportions of inappropriate encodings (OFE, UFE) are high, as in VGG16, DFE can improve both efficiency and performance. However, DFE alone cannot guarantee simultaneous improvement. Thus, we proposed AT and FD, which encourage diverse spike encodings at the cost of efficiency. These methods deteriorate the efficiency improvement due to DFE (increased the number of encoded spikes in Table 4) but increase the diversity of encodings (# of channels in Table 5), which results in improvement of the accuracy. In conclusion, contrary to the reviewer's concern, the effectiveness of our method is not limited to a specific model architecture as in Table 2. The experimental results in Table 4 show the validity of our method, which utilizes complementary mechanisms.
>
> The characteristics of input spike encoding may vary depending on the model architecture. As shown in the baseline of Table 5, VGG16 and ResNet20 exhibit different distributions of encoding channels. We observed that the proportion of improper encodings (OFE, UFE) is lower in the ResNet architecture compared to the VGG architecture. Due to the reasons mentioned above, applying DFE alone improved both performance and efficiency in VGG, whereas in ResNet, only efficiency was improved. When all proposed methods were applied, we observed an improvement in both performance and efficiency compared to the baseline. Therefore, our approach can improve both efficiency and performance, regardless of the various encoding patterns depending on the models.
>
> **Q3. How would DFE perform in networks without Batch Normalization?**
>
> A3. DFE is a method that improves encoding efficiency by adjusting the input distribution of encoding neurons. In this study, we implemented DFE by utilizing the scale and shift parameters of Batch Normalization (BN). As most state-of-the-art deep SNN models currently utilize batch normalization (BN) in the encoding layer, we believe our method is both applicable and practical [Fang et al., 2021; Zheng et al., 2021; Zhou et al., 2023; Yao et al., 2024]. Furthermore, even if BN is not used, the methodology of DFE, regulating spike encoding by adjusting the input distribution of neurons, can still be applied. In such cases, it could be implemented using a KL-divergence loss.

---

> ### Author Response · Authors · 2024-11-22
>
> **Q4. What is the purpose of fitting FD to a probability density function, and which PDF is used? What is the impact of different PDFs?**
>
> A4. For encoding diverse features, we proposed FD loss that increased the diversity of each neuron’s firing in the encoding layer. In order to utilize this as a regularizer during training, in this study, we fitted the distribution with a normal distribution, which is differentiable and can well represent the spike count distribution of neurons. To investigate the effects of other PDFs, we conducted additional experiments using an exponential distribution, and the results are presented in Table R1. For a fair comparison, we report the average of four experiments conducted under the same experimental settings as in the paper. Compared to the baseline, the normal distribution showed an improvement of approximately 0.2%, while the exponential distribution showed an improvement of about 0.08%. The total number of spikes decreased by 4K for the normal distribution and 6K for the exponential distribution, while the encoding spikes decreased by 6K and 7K, respectively. Both distributions outperformed the baseline, but we chose the normal distribution for its superior performance.
>
> Table R1. Comparison of accuracy, number of all spikes, and number of encoded spikes across different PDF (CIFAR10, VGG16).
> | Methods | Distributions | Acc. (%) | # of All Spikes (K) | # of Encoded Spikes (K)|
> |-|-|-|-|-|
> |baseline|-|93.47 ± 0.14|148 ± 8|59 ± 2.0|
> |H-Direct (ours)|normal|93.67 ± 0.06|144 ± 6|52 ± 0.8|
> ||exponential|93.55 ± 0.07|142 ± 7|51 ± 2.0|
>
> **Q5. Does H-Direct show improvements in other architectures, such as RNNs or Transformers?**
>
> A5. The input spike encoding method proposed in this study is model-agnostic. Thus, we believe that it can be applied not only to various CNN models demonstrated in our experiments but also to other architectures such as RNNs and Transformers. For comparison with closely related works, we primarily conducted experiments on CNNs as in [Rathi & Roy, 2021; Zheng et al., 2021; Deng et al., 2022; Guo et al., 2022a; Guo et al., 2023; Mukhoty et al., 2023; Jiang et al., 2024]. Like other related studies, we also thoroughly conducted experiments on various CNN models and datasets in this paper to ensure comprehensive evaluation. Furthermore, to demonstrate the agnostic property of training algorithms, we also performed experiments applying different training algorithms (IM, RMP). This experiment, which has not been shown in other related works, indicates the thorough validation of our methodology. Despite our comprehensive validation, we plan to conduct further validation on other model architectures in the future.

---

> ### Author Response · Authors · 2024-11-22
>
> **Q6. Could the authors analyze the robustness of hyperparameters for the DFE, AT, and FD losses?**
>
> A6. Table R2, R3, and R4 present the accuracy, the number of all spikes, and the number of encoded spikes for different hyperparameters of DFE, AT, and FD, respectively. The hyperparameters used in the paper are indicated in bold. Each experiment was conducted under the same conditions as in our main results, with only the corresponding hyperparameter modified. The results are the average of two trials. First, Table R2 shows the results of experiments conducted by varying the hyperparameter (α) of DFE. When α=−3 compared to α=−1, there is a difference of approximately 0.17% in accuracy, 19K in total spikes, and 8K in encoded spikes. PE tends to have relatively larger  α  values compared to DFE. Thus, as  α  increases, more channels fire in PE, whereas, as  α  decreases, more channels fire in DSE or UFE. Consequently, the number of encoding spikes and total spikes tends to increase as α increases. Next, Table R3 presents the results of experiments by varying the hyperparameter (η) of AT. When η=0.8 compared to η=1.6, there is a difference of approximately 0.32% in accuracy, 20K in total spikes, and 7K in encoded spikes. As η  increases, the rate of threshold reduction for channels that fail to fire in the encoding layer becomes smaller (η ≤ 1 ) or the thresholds exceed their initial values ( η > 1 ). This leads to a decrease in spikes in the encoding layer as  η  increases. The differing trend observed in the total spike count can be attributed to the fact that our method operates only in the encoding layer. Finally, Table R4 shows the results for the hyperparameter (λ_FD) of FD. When λ_FD=5e−7 compared to λ_FD=5e−6, there is a difference of 0.25% in accuracy, 8K in total spikes, and 2K in encoded spikes. FD is a loss function designed to encourage neurons in the encoding layer to fire with diverse spike counts. As  λ_FD increases, the spikes of each neuron are distributed more diversely across the range of 0 to  timestep T . Thus, the encoding spikes tend to increase as λ_FD  increases. Similar to AT, the differing trends in total spikes are likely due to our method being applied only in the encoding layer. We determined each hyperparameter value by considering both performance and efficiency.
>
> Table R2. Comparison of accuracy, number of all spikes, and number of encoded spikes based on the hyperparameter (α) of DFE (CIFAR10, VGG16).
> | α | Acc. (%) | # of All Spikes (K) | # of Encoded Spikes (K)|
> | - | - | - | - |
> | -3 | 93.50 ± 0.07 | 125 ± 3 | 44 ± 3.0 |
> | -2 | 93.50 ± 0.20 | 133 ± 4 | 49 ± 1.0 |
> | **-1** | **93.67 ± 0.06** | **144 ± 6** | **52 ± 0.8** |
> | 1 | 93.60 ± 0.01 | 166 ± 1 | 68 ± 1.0 |
>
> Table R3. Comparison of accuracy, number of all spikes, and number of encoded spikes based on the hyperparameter (η) of AT (CIFAR10, VGG16).
> | η | Acc. (%) | # of All Spikes (K) | # of Encoded Spikes (K)|
> | - | - | - | - |
> | 0.4 | 93.60 ± 0.07 | 136 ± 0.4 | 53 ± 1.0 |
> | **0.8** | **93.67 ± 0.06** | **144 ± 6.0** | **52 ± 0.8** |
> | 1.2 | 93.53 ± 0.13 | 132 ± 5.0 | 51 ± 6.0 |
> | 1.6 | 93.35 ± 0.01 | 124 ± 2.0 | 45 ± 1.0 |
>
> Table R4. Comparison of accuracy, number of all spikes, and number of encoded spikes based on the hyperparameter (λ_FD) of FD (CIFAR10, VGG16).
> | λ_FD | Acc. (%) | # of All Spikes (K) | # of Encoded Spikes (K)|
> | - | - | - | - |
> | 5e-8 | 93.50 ± 0.05 | 135 ± 3.0 | 46 ± 0.04 |
> | 5e-7 | 93.42 ± 0.10 | 136 ± 0.1 | 50 ± 3.00 |
> | **5e-6** | **93.67 ± 0.06** | **144 ± 6.0** | **52 ± 0.80** |
> | 5e-5 | 93.61 ± 0.00 | 142 ± 7.0 | 57 ± 4.00|
> | 5e-4 | 93.47 ± 0.03 | 148 ± 2.0 | 65 ± 5.00 |
>
> Thank you for taking the time to review our work. We hope our answer addresses your concerns. If you have any further questions, please let us know.

---

> > ### Comment · Reviewer_MhWu · 2024-11-25
> >
> > I wonder what's the difference between this paper and Homeostasis-aware Direct Spike Encoding for Deep Spiking Neural Networks https://openreview.net/pdf?id=Uvsa7vFWcT They have a lot of similarities in both images and content. Due to this reason, I will consider lowering my score

---

> ### Author Response · Authors · 2024-11-25
>
> The paper the reviewer mentioned is our work that will be presented at the NeurIPS 2024 workshop (NeuroAI).
>
> As stated in the call for papers for that workshop, papers presented at this workshop are ***not considered for publication (”non-archival”)***.
>
> Call for papers for the workshop (https://neuroai-workshop.github.io/call-for-papers/)
>
> “… We invite researchers in the field of NeuroAI to submit their papers to this workshop. The workshop papers are ***non-archival*** and authors are also encouraged to submit short versions of their NeurIPS main conference submissions concurrently to this workshop. …”
>
> Also, according to ICLR’s dual submission policy, ***presented papers at workshops without publication proceedings do not violate the submission policy***.
>
> Dual Submission Policy of ICLR (https://iclr.cc/Conferences/2025/CallForPapers)
>
> “… However, papers that cite previous related work by the authors and papers that have appeared on non-peer reviewed websites (like arXiv) or ***that have been presented at workshops (i.e., venues that do not have publication proceedings) do not violate the policy***. … ”
>
> Therefore, there is no problem with our submission of this manuscript, and we ask that the reviewers appropriately evaluate this paper with our response above.

---

> > ### Comment · Reviewer_MhWu · 2024-11-26
> >
> > Based on the authors' responses, I have decided not to lower my score. However, I will not raise it either, as the authors did not adequately address my concerns. In my view, their approach lacks sufficient evidence to demonstrate its generalization capabilities, particularly in networks without batch normalization (BN) or architectures like RNNs and Transformers.
> > The primary focus of the paper seems to be addressing incorrect spike firing (e.g., over-firing or under-firing), but this issue is not limited to the encoding layer. As the network deepens, this problem intensifies. Addressing this issue solely in the encoding layer is insufficient and lacks justification. Furthermore, using the first layer as the encoding layer itself is not a well-grounded approach to addressing encoding problems in SNNs. The proposed method does not reasonably solve these challenges.

---

> > > ### Author Response · Authors · 2024-11-28
> > >
> > > Our proposed method, DFE, adjusts the input distribution of neurons by utilizing parameters of Batch Normalization (BN). We believe that similar effects could be achieved using KL divergence. Unlike DFE, methods such as AT and FD are independent of BN parameters. Based on this, we hypothesize that replacing DFE with KL divergence would yield comparable results even in architectures that do not include BN. Furthermore, as an early research that demonstrates the importance of efficient direct encoding, we have rigorously validated our method across various datasets, architectures, and methods. This comprehensive evaluation allows us to reasonably infer the effectiveness of our approach in architectures with similar structures.
> > >
> > > As the reviewer pointed out, we acknowledge that inappropriate neuron firing is not limited to the encoding layer and tends to worsen as the network depth increases. However, the primary goal of our study is to highlight the importance and necessity of efficient input spike encoding in deep SNNs. For this reason, our proposed method has been applied exclusively to the encoding layer. We do not claim that applying our method solely to the encoding layer resolves all issues throughout the entire network. However, we believe our study is significant in demonstrating that improving input encoding alone can lead to substantial enhancements in performance and efficiency. To address the reviewer’s concern regarding the worsening of such issues in deeper networks, we plan to extend our proposed method to the entire network in future research. We anticipate that applying our method across all layers will also result in improved performance and efficiency compared to existing approaches.
> > >
> > > Regarding the concern that using the first layer as an encoding layer may not be an appropriate approach to address the encoding challenges in SNNs, we would like to clarify our position. Our research focuses on analyzing the limitations of existing direct encoding methods and proposes H-direct as a solution. Among various encoding methods, direct encoding is widely used, where the first layer of the network is assigned as the encoding layer to generate spikes from input data. Many state-of-the-art deep SNN models adopt direct encoding and achieve superior performance with fewer timesteps compared to other encoding methods [Rathi & Roy, 2021; zheng et al.,2021; Feng et al., 2022; Guo et al.,2022a; Guo et al., 2023; Zhou et al., 2023]. Our study specifically aims to enhance the efficiency and performance of this commonly used direct encoding approach. We would like to reiterate that our research is meaningful in demonstrating that performance and efficiency improvements can be achieved solely by improving input encoding.

---

### Meta-Review · Area_Chair_gnhs · 2024-12-21

**Metareview:**

This paper introduces a direct spike encoding for spiking neural networks. The main motivation of the authors is to improve the efficiency and stability of the encoding method. The authors introduced the concept of homeostasis to direct encoding and introduced three new components: Dynamic Feature Encoding (DFE) Loss, Adaptive Threshold (AT), and Feature Diversity (FD) Loss. Experimental results show that H-Direct can achieve higher performance and efficiency for SNN.

After the rebuttal stage, 4 reviewers rate 3, 3, 5, and 6, respectively. Most reviewers agree that H-Direct has conducted sufficient experiments and has good research motivation. This is the advantage of H-Direct. However, there are some criticisms. Reviewers N9Cn and ey4J believe that many of the techniques applied in the method have been widely explored in other papers, and the method lacks unique contributions. In addition, reviewers ey4J and KtLe also believe that the performance improvement of the paper is limited compared with previous work. Reviewer MhWu pointed out that the method is highly dependent on the batch norm module to achieve generalization ability. I think these criticisms are reasonable. I suggest that the author further improve the clarity of the paper, the proposed method, and the presentation of experimental results.

One important discussion during the rebuttal period was about the generalization problem. Reviewer MhWu was concerned about the generalization problem and believed that the proposed algorithm lacked sufficient evidence to prove its generalization ability on SNN. In the reply, the author pointed out that other deep learning techniques such as KL divergence can also achieve similar effects. At the same time, it was also acknowledged by the authors that the method may have unexpected situations as the network depth increases. I think the author's efforts in replying to the reviewer MhWu's questions are worthy of recognition. At the same time, I also think that some of Reviewer MhWu's criticisms are still reasonable. The author should consider the potential impact of the proposed algorithm on network performance in depth. This will help the dissemination of the author's proposed method in the SNN community. In addition, the author should improve the article based on the reviewer's comments.

Overall, I think this paper needs further revision. Therefore, the final decision is to reject this paper.

**Additional Comments On Reviewer Discussion:**

Reviewer KtLe finds his/her concerns addressed and raised their ratings to 6. Reviewer MhWu drops the ratings to 5. Reviewers ey4J and N9Cn think the concerns remain and keep the original rating of 3. I agree that there are still some problems with this paper after the rebuttal period.

---

### Decision · Program_Chairs · 2025-01-22

Reject